# Ultrafast self-heating synthesis of robust heterogeneous nanocarbides for high current density hydrogen evolution reaction

Chenyu Li[1,5], Zhijie Wang[2,5], Mingda Liu[1,5], Enze Wang[1], Bolun Wang[1], Longlong Xu[1], Kaili Jiang[3], Shoushan Fan [3], Yinghui Sun [4✉], Jia Li [2✉] & Kai Liu [1✉]

Designing cost-effective and high-efficiency catalysts to electrolyze water is an effective way of producing hydrogen. Practical applications require highly active and stable hydrogen evolution reaction catalysts working at high current densities ($\geq$1000 mA cm$^{-2}$). However, it is challenging to simultaneously enhance the catalytic activity and interface stability of these catalysts. Herein, we report a rapid, energy-saving, and self-heating method to synthesize high-efficiency $Mo_2C$/MoC/carbon nanotube hydrogen evolution reaction catalysts by ultrafast heating and cooling. The experiments and density functional theory calculations reveal that numerous $Mo_2C$/MoC hetero-interfaces offer abundant active sites with a moderate hydrogen adsorption free energy $\Delta G_{H^*}$ (0.02 eV), and strong chemical bonding between the $Mo_2C$/MoC catalysts and carbon nanotube heater/electrode significantly enhances the mechanical stability owing to instantaneous high temperature. As a result, the $Mo_2C$/MoC/carbon nanotube catalyst achieves low overpotentials of 233 and 255 mV at 1000 and 1500 mA cm$^{-2}$ in 1 M KOH, respectively, and the overpotential shows only a slight change after working at 1000 mA cm$^{-2}$ for 14 days, suggesting the excellent activity and stability of the high-current-density hydrogen evolution reaction catalyst. The promising activity, excellent stability, and high productivity of our catalyst can fulfil the demands of hydrogen production in various applications.

---

[1] State Key Laboratory of New Ceramics and Fine Processing, School of Materials Science and Engineering, Tsinghua University, Beijing 100084, China. [2] Shenzhen Geim Graphene Center and Institute of Materials Research, Tsinghua Shenzhen International Graduate School, Tsinghua University, Shenzhen 518055, China. [3] Department of Physics and Tsinghua-Foxconn Nanotechnology Research Center, Tsinghua University, Beijing 100084, China. [4] Beijing Key Laboratory for Magneto-Photoelectrical Composite and Interface Science, School of Mathematics and Physics, University of Science and Technology Beijing, Beijing 100083, China. [5] These authors contributed equally: Chenyu Li, Zhijie Wang, Mingda Liu. ✉email: yhsun@ustb.edu.cn; li.jia@sz.tsinghua.edu.cn; liuk@tsinghua.edu.cn

Hydrogen evolution reaction (HER), which involves overall water splitting, has been considered as a promising approach for hydrogen production[1]. Catalytic activity and stability are important criteria for high-efficiency HER catalysts. In industrial applications, an HER catalyst must be highly active and stable on an electrode at high current densities (e.g., $\geq 500$ or $1000\ mA\ cm^{-2}$) over a long period of time ($\geq 300\ h$)[2]. Pt is usually used for HER but limited for the high cost and scarcity, and thus Pt-group metal-free catalysts with the HER activity comparable to Pt have been extensively studied. However, the development of high-efficiency and Pt-group metal-free HER catalytic electrodes for high-current-density HER is challenging because it requires simultaneous high chemical activity, high chemical stability, and high mechanical stability of the electrodes. In recent years, high activity of HER catalysts at high current densities has been reported via heterogeneous atom doping[3–5], composite synergy[6–10], morphology engineering[11,12], and strain engineering[13]. Nevertheless, many HER catalysts with high activity usually exhibit low chemical stability, and large mechanical forces suffered by active sites during the release of large quantities of $H_2$ bubbles would continually exfoliate the catalyst from the electrodes, making their mechanical stability difficult to maintain at high current densities[14–16].

It is difficult to simultaneously improve the chemical and mechanical stabilities without affecting the activity of an HER catalyst. To improve the chemical stability, chemically stable Pt-group metal-free catalysts, such as $Mo_2C$[17], $MoS_2$[18,19], $MoS_2/Ni_3S_2$[9], and $Cr_{1-x}Mo_xB_2$[20] were explored. To enhance the mechanical stability, robust catalytic electrodes were directly employed[21], or binders were used to strengthen the adhesion between the catalyst and electrode[22]; moreover, specific channels or superaerophobic structures of electrodes were designed to reduce the mechanical forces generated in catalysts by facilitating the release of bubbles[10,12]. However, these approaches to enhancing the stability may weaken the chemical activity of the HER catalysts. For example, additional binders obstruct the exposure of active sites and reduce the overall activity[14,23,24]. Therefore, development of new methods for intrinsically enhancing the stability of HER catalysts and simultaneously maintaining their high activity is crucial for applications at high current densities.

Instantaneously creating chemical bonding between the active catalysts and electrodes may be an effective way to intrinsically improve the mechanical stability of HER catalysts and not influence their chemical activity and stability. In this regard, fast self-heating (Joule-heating) of a conductive matrix could be used to in situ synthesize chemically bonded catalysts on the matrix and avoid the decay of their activity caused by agglomeration, which is superior to traditional methods[25–27]. Herein, we develop a low-energy-consumption method using a carbon nanotube (CNT) film as a heat source and matrix, which rapidly changes its temperature in hundreds of milliseconds to in situ synthesize a robust $Mo_2C/MoC/CNT$ composite film in the presence of Mo and C precursors. The as-prepared uniformly dispersed $Mo_2C/MoC$ heterogeneous nanoparticles are tens of nanometres in size and form strong chemical bonds with the CNT film. Consequently, massive $Mo_2C/MoC$ interfaces offer abundant active sites for HER, resulting in the $Mo_2C/MoC/CNT$ film with a low overpotential of 255 mV at a high current density of $1500\ mA\ cm^{-2}$ in 1 M KOH. The strong chemical bonds between $Mo_2C/MoC$ and CNTs significantly weaken the dissolution and shedding of the $Mo_2C/MoC$ nanoparticles during the HER at high current densities. As a result, the overpotential of the $Mo_2C/MoC/CNT$ film changes by only ~32 and ~47 mV after working at 500 and $1000\ mA\ cm^{-2}$ for 14 days, respectively. Density functional theory (DFT) calculations demonstrate the moderate free energy ($\Delta G_{H*}$) of 0.02 eV for hydrogen adsorption at sites around $Mo_2C/MoC$ interfaces and a strong coupling between the $Mo_xC$ and CNT matrix, which ensures the high activity and stability of the heterogeneous $Mo_2C/MoC/CNT$ film.

## Results

**Self-heating synthesis of $Mo_2C/MoC/CNT$ catalysts**. The synthesis of $Mo_2C/MoC/CNT$ catalysts by self-heating is shown in Fig. 1a. First, precursors including ammonium molybdate as metal source and glucose and urea as carbon source were loaded onto a laser-drilled CNT film[10] by dip coating (step 1). Second, the CNT film loaded with precursors was dried at 60 °C for 10 min in air (step 2). Finally, self-heating synthesis was performed in a mixed atmosphere of 10% $H_2$ and 90% Ar with a total flow rate of 200 sccm, in which the precursors in situ reacted on the rapidly Joule-heated CNT film (step 3), resulting in a $Mo_2C/MoC/CNT$ composite film as an HER electrode (Fig. 1b). It took approximately ~256 ms to heat up from room temperature to ~1770 K and ~330 ms to cool down from ~1770 K to ~600 K (Fig. 1c and Supplementary Fig. 1a). The CNT films emit visible light after the application of a voltage of ~ 0.5 V/mm and show a uniform distribution of temperature (Fig. 1d and inset). The as-prepared $Mo_2C/MoC/CNT$ film remains flexible, akin to the pure CNT, MoC/CNT, and $Mo_2C/CNT$ films before and after the rapid treatment at ~1770 K (Supplementary Fig. 2). We simulated the in-plane temperature distribution of the CNT film at ~1700 K and the central temperature under the same power density with different CNT film sizes (by fixing the aspect ratio at 15:8) using the COMSOL Multi-Physics software. The constant temperature zone increases with the size of the CNT film and is adjustable to the application requirements (Fig. 1e and Supplementary Fig. 1b, d), although the temperature rapidly changes within ~1 mm at the boundary in contact with the graphite clips. This method is also suitable for large-scale fabrication (inset of Fig. 1e).

**Structural characterizations**. Raman spectroscopy and X-ray diffraction (XRD) spectroscopy were performed to analyse the chemical composition of the $Mo_2C/MoC/CNT$ film. The $Mo_2C/MoC/CNT$ film exhibits four apparent Raman peaks, of which two peaks at 822 and $995\ cm^{-1}$ belong to $\beta$-$Mo_2C$, and two peaks at ~1350 and ~$1580\ cm^{-1}$ belong to CNT (Fig. 2a)[28–30]. As shown in Fig. 2b and Supplementary Fig. 3a, the strong XRD peaks originate from $\beta$-$Mo_2C$, $\alpha$-MoC, and CNT. The fitted weight percentages of $\alpha$-MoC and $\beta$-$Mo_2C$ are 59.8% and 40.2%, respectively, corresponding to an $\alpha$-MoC-to-$\beta$-$Mo_2C$ molar ratio of about 2.8:1. Besides these peaks, there is a weak XRD peak at about $2\theta = 11.5°$ (Supplementary Fig. 3b), corresponding to a reduced c lattice parameter of a transition phase from 2D $Mo_2C$ (MXene) to $\beta$-$Mo_2C$. This may result from the defunctionalization of the surface terminations, the removal of intercalated water, and the reestablishment of long-range order in 2D $Mo_2C$[31,32]. These results indicate that $\beta$-$Mo_2C$ and $\alpha$-MoC are major phases in the composite and the transition phase from 2D $Mo_2C$ to $\beta$-$Mo_2C$ is minor. Furthermore, we qualitatively studied the trend of component changes through the peak intensity in the XRD patterns. The ratio of ammonium molybdate to glucose in the precursor solution adjusts the final loading content of $Mo_2C$ and MoC as well as the ratio of MoC to $Mo_2C$. As shown in Fig. 2c and Supplementary Fig. 4, the heating time at 135 W also regulates $MoC:Mo_2C$. The XRD peak intensity ratio of MoC (111) to $Mo_2C$ (002) decreases with increased heating time, which indicates an increased content of $Mo_2C$ or a decreased content of MoC and verifies the conversion of MoC to $Mo_2C$ at high temperatures. When the heating time is long enough, the $Mo_2C$ phase completely replaces the hybrid $Mo_2C/MoC$, suggesting that a short reaction time is crucial for synthesizing the $Mo_2C/MoC$ heterostructure (Supplementary Fig. 5 and Supplementary Fig. 6).

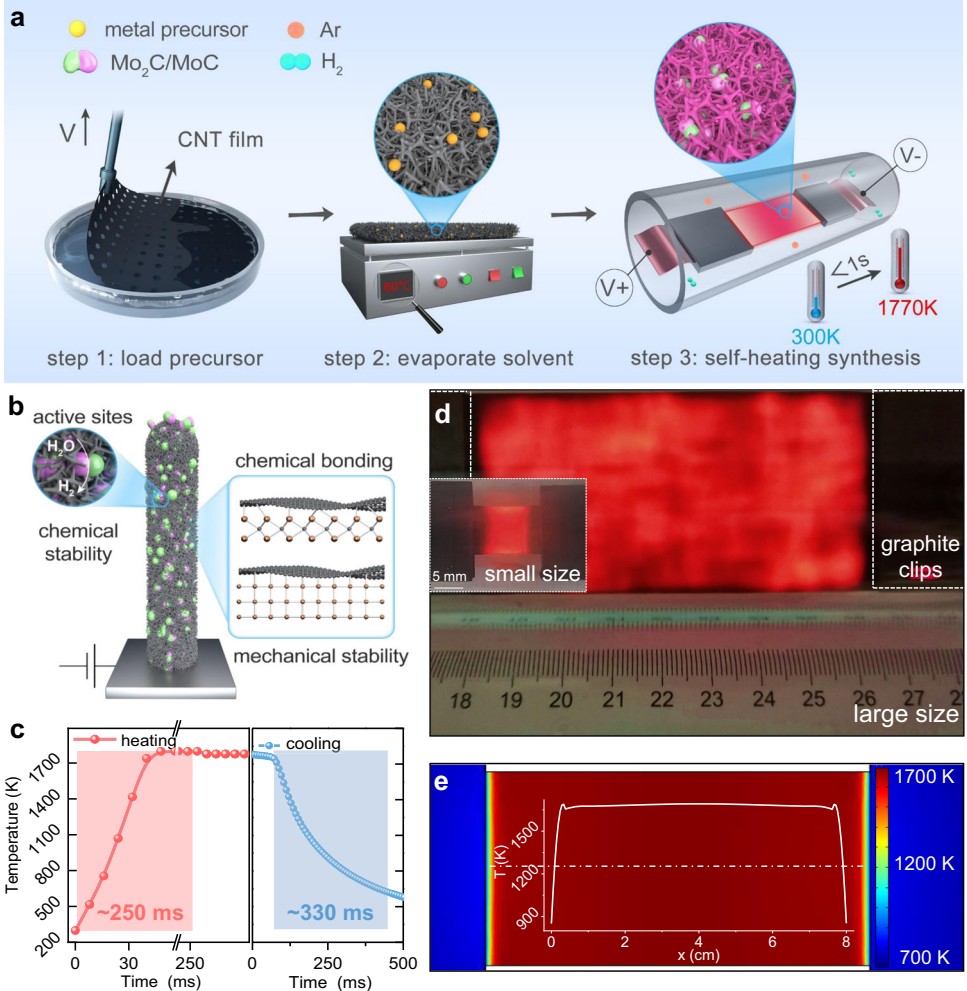

**Fig. 1 Ultrafast self-heating synthesis. a** Schematic illustration of the synthesis of Mo₂C/MoC/CNT films by the self-heating method. **b** Mo₂C/MoC/CNT film as an HER electrode. **c** Temperature-Time curve of heating (left) and cooling (right) process during self-heating. **d** Optical pictures of the CNT film at high temperature in a large size (80 mm × 40 mm) and a small size (15 mm × 8 mm, inset). **e** Simulation of temperature distribution (colour contour) at ~1700 K via COMSOL Multi-Physics software for the large-size film. The top-layer curve shows the temperature distribution along the centerline.

X-ray photoelectron spectroscopy (XPS) was performed to analyse the electronic states. As shown in Supplementary Fig. 7a, clear Mo and C peaks appeared in the full spectra. The peaks of Mo 3d are divided into three groups, including two peaks of $Mo^{2+}$ at 228.5 and 231.7 eV, two peaks of $Mo^{3+}$ at 228.9 and 232.1 V, and two peaks of $Mo^{4+}$ at 229.9 and 233.1 eV, respectively (Fig. 2d)[33–35]. It is still debating for the exact valence states of Mo in Mo₂C or MoC. Most studies suggest the dominance of $Mo^{2+}$ in Mo₂C and $Mo^{3+}$ in MoC, and higher valence states of $Mo^{4+}$, $Mo^{5+}$, and $Mo^{6+}$ result from partial oxidation[36,37]. Under this assumption, compared with the $Mo^{2+}$ peak in Mo₂C and the $Mo^{3+}$ peak in MoC, in the Mo₂C/MoC/CNT film, the $Mo^{2+}$ peak apparently blueshifts (~0.6 eV) while the $Mo^{3+}$ peak redshifts (~0.5 eV), suggesting the existence of electron transfer from Mo₂C to MoC in the heterogeneous composite. Note that the strong $Mo^{4+}$, $Mo^{5+}$, and $Mo^{6+}$ peaks in the MoC sample should result from the surface $MoO_x$ because MoC is very prone to oxidation. Furthermore, we investigated the charge distribution in Mo₂C and MoC by DFT calculations. As shown in Supplementary Fig. 8, each Mo atom loses about 0.80 and 0.50 electron, respectively, in the body of the MoC (111) and Mo₂C (100) regions, while at the Mo₂C/MoC interface, each Mo atom loses about 0.75 and 0.70 electron because of the

electron transfer from Mo₂C to MoC regions. This result clearly proves that the charge transfer exists solely at the Mo₂C/MoC interface. In addition, evident peaks in the C 1 s spectra of C–C, C–O, O = C–O, and Mo–C are derived from CNT and $Mo_xC$ (Fig. 2e)[38]. However, the peaks of N 1 s are not observed in the XPS spectra though urea is a precursor. This may be because N cannot replace C as a doping atom at a high temperature of ~1770 K (Supplementary Fig. 7b) during rapid heating. We also verified the absence of N using a material element analyser, as shown in Supplementary Table. 1. Thermogravimetric (TG) analysis was performed in air at 30–700 °C to estimate the content of each component in the Mo₂C/MoC/CNT film (Fig. 2f). A significant increase in weight from ~250 to ~450 °C is mainly due to the oxidation of MoC and Mo₂C to MoO₃, followed by a weight loss caused by the oxidation of CNTs to CO₂. Combining XRD and TGA data, the wt% of Mo₂C, MoC, and CNTs in the Mo₂C/MoC/CNT film is estimated to be ~15.4%, 22.8%, and 61.8%, respectively.

The surface morphology of the Mo₂C/MoC/CNT film was observed using scanning electron microscopy (SEM). To improve the electrode stability, the CNT film was drilled by a laser to construct periodic micropores as the channels for H₂ release (Supplementary Fig. 9)[10]. As shown in Fig. 3a, b the

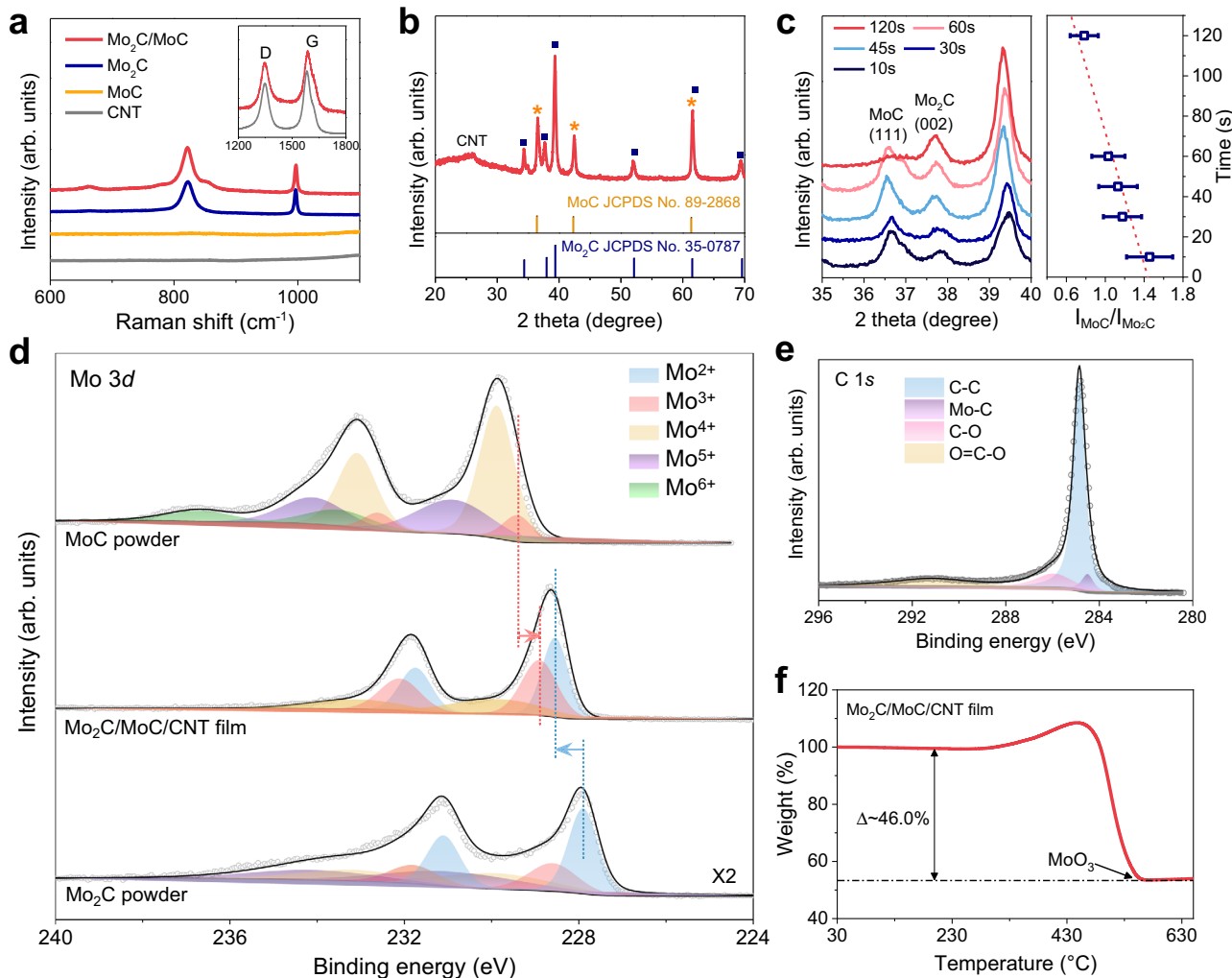

**Fig. 2 Structural characterizations of as-synthesized Mo₂C/MoC/CNT films. a** Raman spectra of Mo₂C/MoC/CNT film, Mo₂C powder, MoC/CNT film, and CNT film. The inset shows D and G Raman peaks of CNTs. **b** XRD pattern of Mo₂C/MoC/CNT film. **c** XRD spectra of Mo₂C/MoC/CNT films synthesized with different heating time at 135 W (left panel) and the corresponding peak intensity ratios of MoC (111) to Mo₂C (002) (right panel). The error bars represent standard deviations. **d** XPS spectra of Mo 3$d$ in MoC powder (top), Mo₂C/MoC/CNT film (middle), and Mo₂C powder (bottom). **e** XPS spectra of C 1$s$ in the Mo₂C/MoC/CNT film. **f** TG analysis of Mo₂C/MoC/CNT film in air.

catalyst particles are uniformly dispersed on the surface of the CNT film without visible agglomeration. Mo₂C/MoC particles, functioning as solder joints, bind several CNTs together to form a bundle, which tightly connect adjacent CNTs and prevent CNTs from sliding, greatly enhancing the binding force between adjacent CNTs and strengthening the CNT matrix. The small size (~20 nm) and distribution states of Mo₂C/MoC particles facilitate the effective strengthening (Fig. 3c). We also changed the heating ramp time for both heating steps from room temperature to ~1100 K (30 W) and from ~1100 to ~1770 K (135 W). As the heating ramp rate decreases, the Mo₂C/MoC particles gradually agglomerate and increase in size because they are more likely to diffuse and aggregate during the heating process, especially at high temperatures (Supplementary Fig. 10).

Distinct interfaces and two Mo-C phases in the particles were observed by high-resolution transmission electron microscopy (HRTEM, Fig. 3d). The (100) crystal plane of Mo₂C forms an interface with the (111) crystal plane of MoC, whose interplanar distances of ~0.260 and ~0.246 nm are relatively similar. To further detect the phase structure and interface, high-angle annular dark field scanning transmission electron microscopy

(HAADF-STEM) was performed, displaying Mo atoms because of a large Z value. Composite phases of Mo₂C/MoC are formed in most catalytic particles and one typical particle is shown in Fig. 3e, where Mo₂C, MoC, and an interfacial transition zone simultaneously exist. The inset (top right) of Fig. 3e and Supplementary Fig. 11a show the crystal structure of MoC and the atomic sites of Mo, where the angle of (111) and (220) is 90°, which is consistent with the measurement results in HAADF-STEM images and Fourier transform (FT) patterns. The inset (bottom right) of Fig. 3e and Supplementary Fig. 11b show the crystal structure of Mo₂C, where the angles of (100) and (002) are 90° from the measurements, which is consistent with the theoretical prediction. Figure 3f shows the results of selected electron area diffraction (SAED), in which the diffraction spots/circles of Mo₂C (101), MoC (111), and CNT (002) suggest the composition of Mo₂C, MoC, and CNT. Energy-dispersive spectroscopy (EDS) mapping images show the distribution of Mo, C, and N in Fig. 3g and Supplementary Fig. 12. Mo atoms are concentrated on the catalyst particles, and C atoms are concentrated on catalyst particles and CNTs. However, no clear signal of N atoms was detected, which is consistent with the XPS spectra.

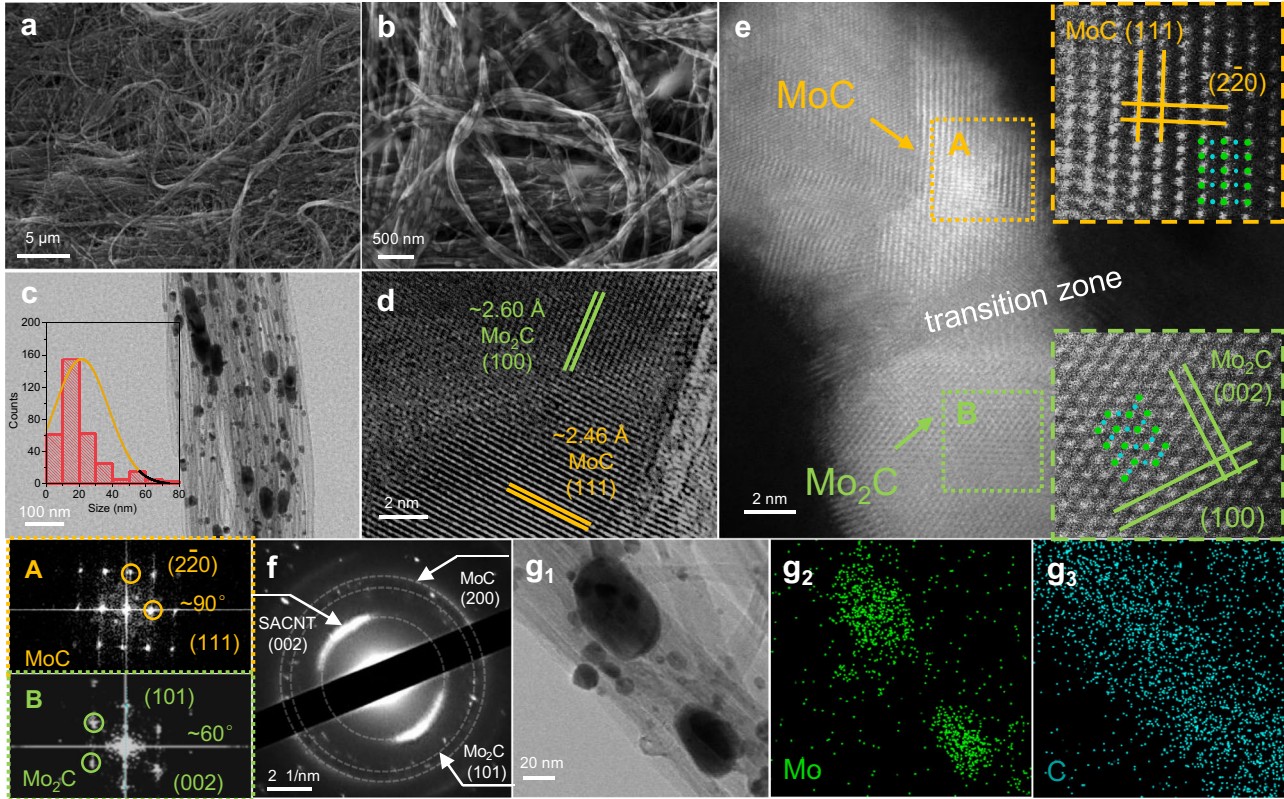

**Fig. 3 Electron microscopic characterizations of Mo$_2$C/MoC/CNT films. a, b** SEM images of Mo$_2$C/MoC/CNT film. **c** TEM images of Mo$_2$C/MoC/CNT film. The inset is the statistics of Mo$_2$C/MoC particle size distribution. **d** HRTEM images of the interface between Mo$_2$C (100) and MoC (111) in Mo$_2$C/MoC/CNT film. **e** HAADF-STEM image of Mo$_2$C/MoC. The inset is HAADF-STEM image of MoC phase and Mo$_2$C phase, as well as the FT patterns. Green dots represent Mo atoms, and blue dots represent C atoms. **f** SAED pattern of Mo$_2$C/MoC/CNT film. **g** EDS mapping shows the distribution of Mo and C elements.

**Electrochemical performance at high current densities**. The electrochemical performance of the catalysts was measured in 1 M KOH (pH = 14) in a three-electrode configuration. Different Mo$_2$C/MoC/CNT samples were prepared by changing the type of carbon sources, e.g. glucose, sodium phenolate, and phenylalanine, the content of the precursor, the weight percent of CNT, and the heating time (Supplementary Figs. 13–18). Glucose is a suitable carbon source owing to its high activity, easily breaking C-C bonds and forming Mo-C bonds within a short time (Supplementary Fig. 13). Considering the ratio of Mo to C in the precursor, an insufficient C source cannot react with the Mo source on time before the Mo source is evaporated out, leading to a small Mo$_2$C/MoC loading content and insufficient Mo$_2$C content. However, excess C source may cover Mo$_2$C/MoC, preventing the active sites at Mo$_2$C/MoC from thorough exposure (Supplementary Fig. 14). In our method, the light content of urea fine-tunes the atomic ratio of Mo to C, and the fast self-heating process adjusts the conversion from MoC to Mo$_2$C, which further optimizes the ratio of MoC to Mo$_2$C (Supplementary Figs. 15 and 16). A suitable proportion of MoC and Mo$_2$C offers richer interfaces and provides more HER active sites. The optimal ratio of Mo to C source and glucose to urea are 4:3 and 20:3, respectively. Besides the ratios of Mo to C source and glucose to urea, the CNT wt% also plays a great role in the performance of the Mo$_2$C/MoC/CNT films. As shown in Supplementary Fig. 17, the sample with the moderate CNT wt% (~61.8 wt% CNT and ~38.2 wt% Mo$_2$C/MoC) possesses the best HER activity because it exhibits no apparent agglomeration of nanoparticles and thus has abundant Mo$_2$C/MoC interfacial area. More or less content of Mo$_2$C/MoC will lead to poorer HER activity. When the content of

CNT is high, the content of MoC or Mo$_2$C is very few, and thereby the total Mo$_2$C/MoC interfacial area becomes very limited. And for the samples with a low content of CNT, Mo$_2$C/MoC will agglomerate during the self-heating process, also reducing the total Mo$_2$C/MoC interfacial area (Supplementary Fig. 18). This result suggests that an appropriate CNT wt% is needed to increase the Mo$_2$C/MoC interfacial area for a better HER.

As shown in Fig. 4a, the as-prepared Mo$_2$C/MoC/CNT film requires the overpotentials of only 82, 201, 233, and 255 mV to achieve HER current densities of 10, 500, 1000, and 1500 mA cm$^{-2}$, respectively. Compared to precious metal electrodes, Mo$_2$C/MoC/CNT films perform better at current densities over ~450 mA cm$^{-2}$ than Pt/C, which is considered as a benchmark for HER catalysts. The overpotential of the Mo$_2$C/MoC/CNT film is notably smaller than that of the Mo$_2$C/CNT, MoC/CNT, and CNT films throughout the entire HER process, for which the overpotentials are ~318, >350, and >500 mV at 1000 mA cm$^{-2}$, respectively. Compared to the Mo$_2$C/CNT film, the MoC/CNT film, and especially the physically mixed Mo$_2$C/MoC/CNT film (m-Mo$_2$C/MoC/CNT film, Supplementary Fig. 19), the superiority of the Mo$_2$C/MoC/CNT film made by self-heating suggests the importance of the Mo$_2$C/MoC interface, where Mo is more conducive to the adsorption and desorption of H* via interfacial charge transfer[33]. Otherwise, either strong adsorption or strong desorption of H* weaken the activity of HER, leading to the lower electrochemical activity of the MoC/CNT and Mo$_2$C/CNT films, respectively. To investigate the kinetics, Tafel slopes were calculated from the polarization curves (Supplementary Fig. 20). The Tafel slope of the Mo$_2$C/MoC/CNT film is 42 mV dec$^{-1}$, which is smaller than that of the Mo$_2$C/CNT film (50 mV dec$^{-1}$), MoC/CNT film

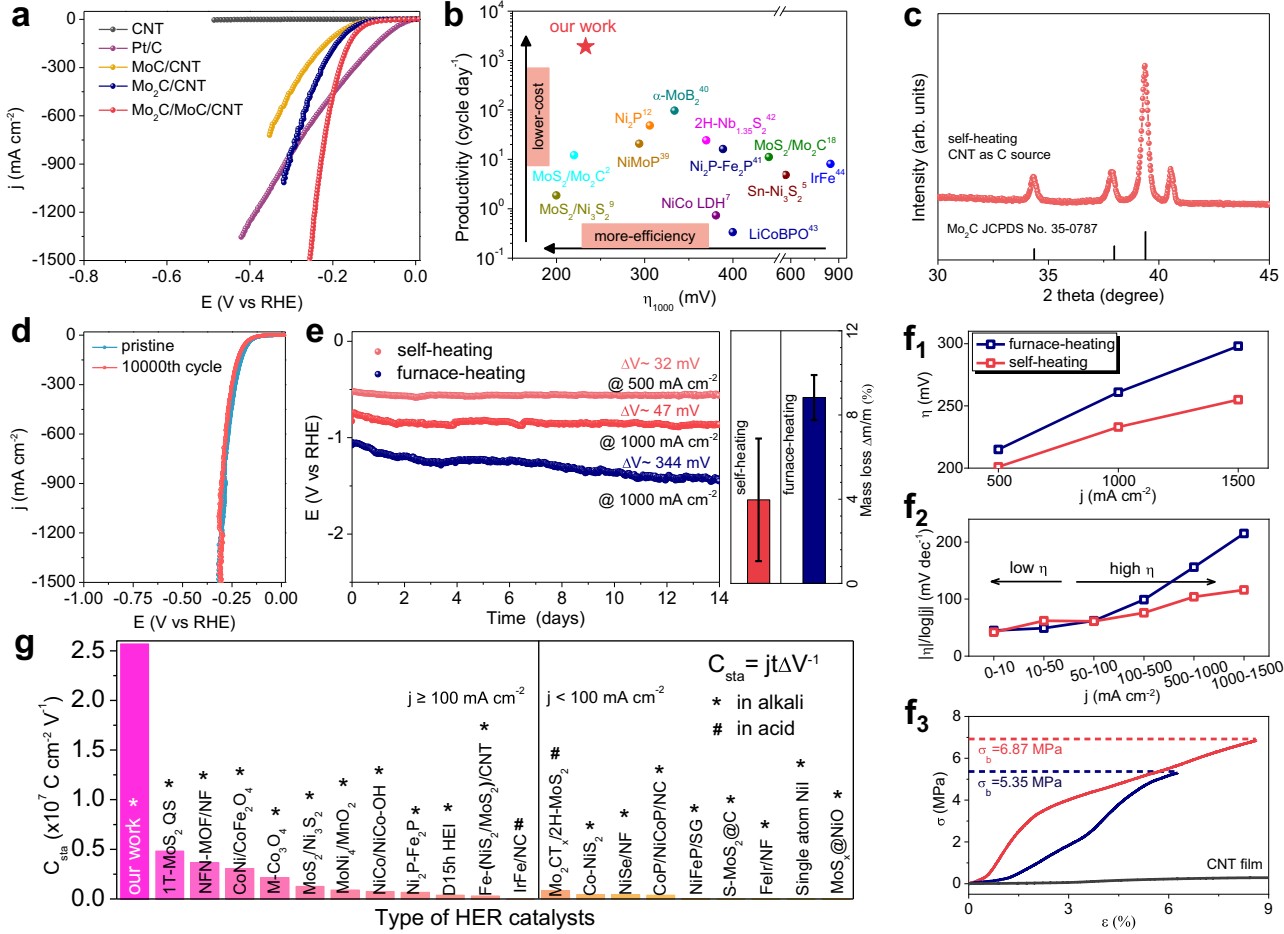

**Fig. 4 Electrocatalytic HER performance of Mo₂C/MoC/CNT films. a** Polarization curves of Mo₂C/MoC/CNT film, Mo₂C/CNT film, MoC/CNT film, pure CNT film and Pt/C (20 wt% of Pt) in 1 M KOH. **b** Comparison of overpotentials at 1000 mA cm⁻² versus productivity (synthesis cycles in a day) among HER catalysts maintaining activity at high current density. **c** XRD patterns of self-heating samples only use CNT as carbon sources. **d** Polarization curves of a pristine Mo₂C/MoC/CNT film and working after 10000 CV cycles. **e** Long-term test of self-heating and furnace-heating samples at 500 or 1000 mA cm⁻² without iR compensation (left). A mass loss of self-heating and furnace-heating sample after 5 min of ultrasonic treatment in 50 mL alcohol (right). The error bars represent standard deviations. **f₁** The overpotential of self-heating and furnace-heating samples at 500, 1000, and 1500 mA cm⁻² in 1 M KOH, respectively. **f₂** The ratios of Δη/Δlog|j| for self-heating and furnace-heating samples in different current density ranges. **f₃** The mechanical tensile curves of self-heating sample, furnace-heating sample, and pure CNT film. **g** Comparison of long-term stability of various HER catalysts at both small and high current densities.

(58 mV dec⁻¹), and is close to that of Pt/C (31 mV dec⁻¹), indicating that HER is based on the Volmer-Heyrovsky mechanism. As the kinetic activity is affected by the electrochemical surface area (ECSA) and charge transfer resistance, we measured the electrochemical double-layer capacitances ($C_{dl}$) of various electrodes, which were proportional to the ECSAs, and charge transfer resistances. As shown in Supplementary Figs. 21 and 22, the Mo₂C/MoC/CNT film exhibits a high $C_{dl}$ of 119.9 mF cm⁻² and a small charge transfer resistance of ~3.2 Ω, suggesting an evident advantage compared to the Mo₂C/CNT (83.5 mF cm⁻² and ~4.5 Ω) and MoC/CNT (66.0 mF cm⁻² and ~6.9 Ω) films. The polarization curves of the three kinds of films normalized to the ECSA and the mass of active materials are shown in Supplementary Fig. 23 ($C_s$ is set to be 40 μF cm⁻²), which demonstrate that the activity of Mo₂C/MoC/CNT film does increase intrinsically compared with those of Mo₂C/CNT film and MoC/CNT film. For $C_{dl}$, the measurement error includes systematic errors (0.01% for voltage application and 0.2% for current detection) and random error (~5%), which is negligible. For $C_s$, as the actual value cannot be determined precisely in this work. $C_s$ for a flat surface is generally found to be in a range of 20–60 μF cm⁻², and the value of

40 μF cm⁻² is used in this work to calculate the ECSA. The actual value of $C_s$ could be very different from the used value and thus an error may be introduced in the absolute value of ECSA. However, the relative values of the ECSA for the materials reported in this work are not affected by the absolute errors because the Mo₂C/MoC/CNT film, Mo₂C/CNT film, and MoC/CNT film are prepared on the same supporting material, the CNT films. The $C_s$ values for all these materials should be nearly identical in principle, and thus the calculated ECSAs can be compared relatively. Moreover, we also evaluated the turnover frequency (TOF) of each catalyst film (Supplementary Fig. 24). At an overpotential of 250 mV, the TOF of Mo₂C/MoC/CNT film, Mo₂C/CNT film, and MoC/CNT film is 0.65, 0.30, and 0.22 s⁻¹, respectively, which validates that Mo₂C/MoC/CNT catalyst has higher intrinsic activity besides the larger ECSA. The increase of the intrinsic activity should be attributed to the increase in the number of Mo₂C/MoC interfaces.

Besides the excellent HER activity of the Mo₂C/MoC/CNT film at high current densities, our self-heating method has notable advantages in the rapid synthesis process and high productivity, which takes a comparable pre-/post-synthesis processing time and a much shorter synthesis time than traditional methods

(Fig. 4b and Supplementary Table. 2). The $\eta_{1000}$ of the $Mo_2C/MoC/CNT$ film is close to $MoS_2/Mo_2C[2]$, $MoS_2/Ni_3S_2[9]$, and is much lower than $Ni_2P/NF[12]$, $Ni_{2(1-x)}Mo_{2x}P-NF[39]$, $\alpha-MoB_2[40]$, NiCo LDH/NF[7], $Ni_2P-Fe_2P[41]$, $2H-Nb_{1.35}S_2[42]$, $HC-MoS_2/Mo_2C[18]$, $Sn-Ni_3S_2[5]$, LiCoBPO/NF[43], and IrFe/NC[44]. The total growth time of one-cycle self-heating synthesis is only 45 s, which means that 1920 synthesis cycles can be theoretically completed within a day, more efficient than traditional furnace heating or solvothermal methods that require several hours for one cycle. Therefore, our self-heating synthesis not only delivers $Mo_2C/MoC/CNT$ films that have excellent HER activity and tolerate high current densities but also achieves high productivity beyond traditional methods.

To study the formation of chemical bonds between the CNTs and $Mo_xC$, we used only CNTs as the carbon source to conduct an experiment under the same reaction conditions without glucose and urea. The XRD patterns show the successful synthesis of $Mo_xC$ (Fig. 4c), indicating that CNTs are involved in chemical reactions and Mo-C bonds are formed. Thus, the interaction between CNTs and $Mo_xC$ is strengthened because of the Mo-C bonds, which significantly improve the electron transfer and stability. As a result, the polarization curve of the $Mo_2C/MoC/CNT$ film after 10,000 CV cycles remains almost unchanged compared with the original polarization curve, with a deviation of only 10 mV from 0 to 1500 mA cm$^{-2}$ (Fig. 4d), which suggests that the $Mo_2C/MoC/CNT$ film is highly stable under CV cycle conditions. As shown in Supplementary Table. 3, the CV cycle stability of the $Mo_2C/MoC/CNT$ film is higher than all of the previous catalysts. We also performed 14-day chronopotentiometry (CP, at fixed current) at large current densities of 500 and 1000 mA cm$^{-2}$ to further verify the stability (Fig. 4e, left). The $Mo_2C/MoC/CNT$ film synthesized by self-heating shows an increase of overpotential by only ~32 and ~47 mV after 14 days (336 h) at 500 and 1000 mA cm$^{-2}$, respectively. In contrast, the overpotential of $Mo_2C/MoC/CNT$ film synthesized by tube furnace heating (denoted as f-$Mo_2C/MoC/CNT$ film, Supplementary Fig. 25) increases by ~344 mV after working at 1000 mA cm$^{-2}$ for 14 days, which is much inferior to that of the $Mo_2C/MoC/CNT$ film by self-heating. The long-term stability of the $Mo_2C/MoC/CNT$ film at 3000 mA cm$^{-2}$ is also much better than that of the f-$Mo_2C/MoC/CNT$ film (Supplementary Fig. 26).

It is found that the HER performance of the $Mo_2C/MoC/CNT$ catalyst was improved at the initial several hours (Supplementary Fig. 27), which may result from the fact that the surface of the as-prepared samples was reconstructed owing to the adsorption of oxygen-containing groups during the HER[2,45]. After a long-term test at a high current density, the content of Pt in the electrolyte is below the detection limit of inductively coupled plasma (ICP) mass spectrometry (Supplementary Table. 4), which excludes the influence of Pt dissolution from counter electrodes during the electrochemical measurements. A similar long-term test at the high current density using a graphite counter electrode also exhibits high stability until the graphite electrode is dissolved (Supplementary Fig. 28). We also used a graphite rod as the counter electrode to measure the CV curves of $Mo_2C/MoC/CNT$ films for 50 cycles, and then changed to use a Pt counter electrode for the other 50-cycles CV measurement. The CV curves obtained by graphite and Pt counter electrodes are almost identical (Supplementary Fig. 29). To further investigate the higher stability of the $Mo_2C/MoC/CNT$ catalyst beyond the f-$Mo_2C/MoC/CNT$ catalyst, we used an ultrasonic machine to treat the catalysts. As shown in Fig. 4e (right), the mass loss of the $Mo_2C/MoC/CNT$ film ($3.2 \pm 2.9\%$) after the ultrasonic treatment is lower than that of the f-$Mo_2C/MoC/CNT$ film ($8.7 \pm 1.1\%$), suggesting a stronger binding between $Mo_2C/MoC$ and CNTs in

the $Mo_2C/MoC/CNT$ film. On the microscopic level, unlike bulk $Mo_2C/MoC$ with poor crystallinity covering CNTs in the f-$Mo_2C/MoC/CNT$ film, the well-crystalline $Mo_2C/MoC$ nanoparticles disperse in the $Mo_2C/MoC/CNT$ film (Supplementary Figs. 30 and 31), leading to huge effective contact areas and sufficient binding strength at the interfaces due to chemical Mo-C bonding. Such strengthened interfaces are sufficient to prevent the $Mo_2C/MoC$ from being peeled off by large quantities of $H_2$ bubbles when working at high current densities. After working for ~6 days at 1000 mA cm$^{-2}$, the $Mo_2C/MoC/CNT$ film changes slightly in the microscopic morphology of $Mo_2C/MoC$ particles, while the f-$Mo_2C/MoC/CNT$ film only has a few agglomerated $Mo_2C/MoC$ remained surrounding the CNTs (Supplementary Fig. 32). The XRD spectrum of the $Mo_2C/MoC/CNT$ film after working at 1000 mA cm$^{-2}$ for ~6 days still shows clear MoC and $Mo_2C$ peaks as well as unnoticeable $MoO_2$ or $MoO_3$ peaks, although the ratio of the peak intensity of MoC to $Mo_2C$ changes (Supplementary Fig. 33) owing to the different etching and shedding rates of the two phases. Although the material is corroded, the remained excellent HER performance indicates that the $Mo_2C/MoC$ interface, rather than single MoC or $Mo_2C$, is crucial in HER. In contrast, XRD peaks belonging to $MoO_2$ and $MoO_3$ appear in the f-$Mo_2C/MoC/CNT$ film after the test in the same period. This indicates that the f-$Mo_2C/MoC/CNT$ film is more prone to oxidation because of its poorer crystallinity, because in the furnace-heating, it is difficult to obtain high crystallinity of the f-$Mo_2C/MoC/CNT$ film and simultaneously maintain the $Mo_2C/MoC$ composite phase due to the much longer heating process at a lower temperature.

As shown in Fig. 4f$_1$, the $\eta_{500}$, $\eta_{1000}$, and $\eta_{1500}$ of the $Mo_2C/MoC/CNT$ film are lower than those of the f-$Mo_2C/MoC/CNT$ film by 14, 28, and 43 mV, respectively. Throughout HER, the Tafel slope of the $Mo_2C/MoC/CNT$ film increases slightly and is below 120 mV dec$^{-1}$, indicating a rapid kinetic process even at a high current density. In contrast, the Tafel slope of the f-$Mo_2C/MoC/CNT$ film increases visibly and exceeds that of the $Mo_2C/MoC/CNT$ film, suggesting a higher reaction resistance (Fig. 4f$_2$). Owing to the good dispersion of the catalyst particles and their tight combination with the CNT matrix, the self-heating samples exhibit a larger $C_{dl}$ and smaller charge transfer resistance than that of the furnace heating ones (94.3 mF cm$^{-2}$ and ~10.4 Ω). Therefore, the self-heating samples are preferable for HER (Supplementary Figs. 34 and 35). Direct mechanical tensile experiments visually illustrate that the $Mo_2C/MoC/CNT$ film has much higher Young's modulus (E = 231.60 MPa), breaking strength ($\sigma_b$ = 6.87 MPa), and toughness (26.16 N m$^{-3}$) than pure CNT film and f-$Mo_2C/MoC/CNT$ film. After the self-heating, the breaking strain of the pure CNT film is notably reduced while the breaking strength only slightly increases to ~0.32 MPa (Supplementary Fig. 36). In contrast, the $Mo_2C/MoC/CNT$ film synthesized by the self-heating process has a much higher breaking strength. These results suggest that the enhancement of the mechanical strength of the $Mo_2C/MoC/CNT$ film should be mostly attributed to the strong interaction between $Mo_2C/MoC$ and CNTs, which improves the load transfer efficiency inside the film.

We have enumerated the long-term stability of the $Mo_2C/MoC/CNT$ film and other HER catalysts. For a quantitative comparison, we introduce a parameter, $C_{sta}$, to evaluate the stability, which represents the discharge of HER per unit area resulting from per millivolt of overpotential change and can be described by a function,

$$C_{sta} = jt\Delta V^{-1} \quad (1)$$

where j is current density, t is the whole working time, and $\Delta V$ is the variation of overpotential. A larger $C_{sta}$ indicates that

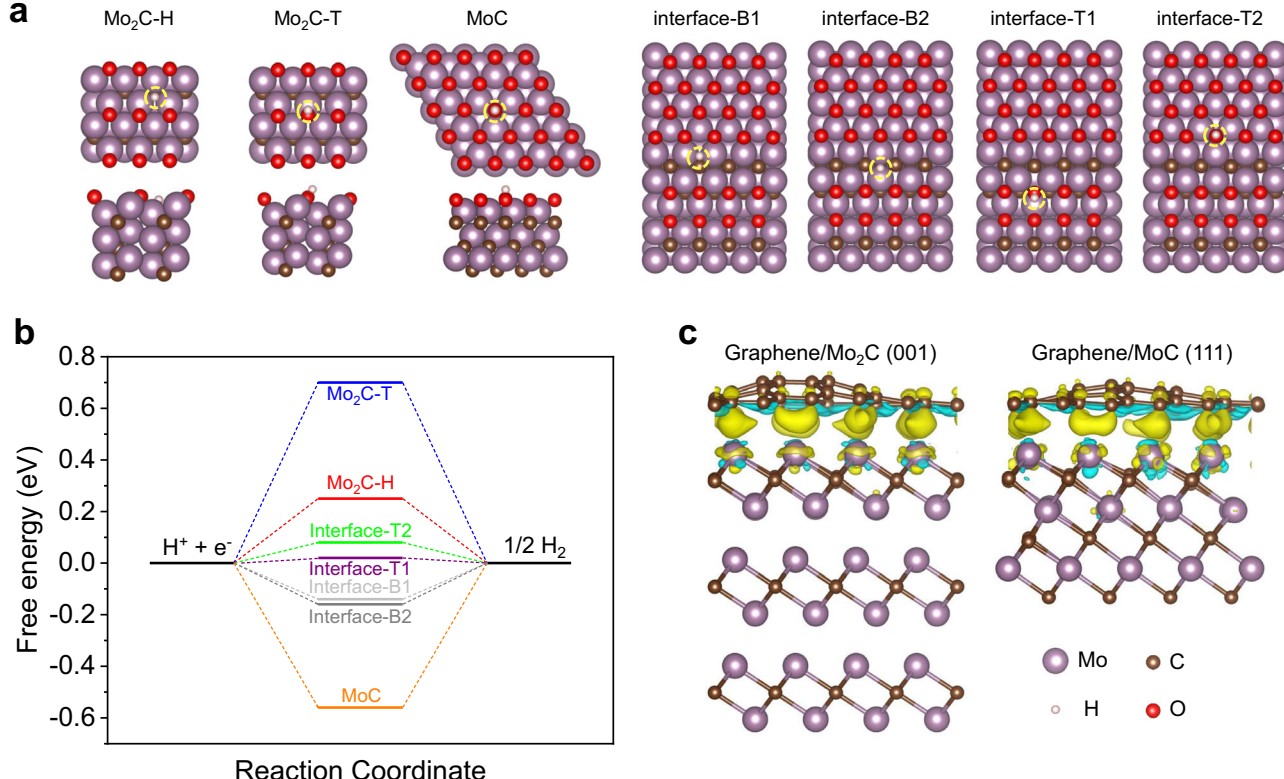

**Fig. 5 DFT calculations. a** Adsorption structures of hydrogen on the Mo₂C (100), MoC (111) surfaces, and Mo₂C (100)/MoC (111) heterostructure, in which the adsorption sites of hydrogen are shown in the yellow dotted circle. The cyan, brown, red and white spheres present the Mo, C, O, and H atoms, respectively (B: bridge site; T: top site; H: hollow site). **b** Free energy diagram of the HER for different adsorption sites on Mo₂C (100), MoC (111), and Mo₂C (100)/MoC (111) heterostructure. **c** Charge density difference of graphene/Mo₂C (001) and graphene/MoC (111) heterostructure. The cyan and yellow regions represent charge depletion and accumulation, respectively, and the isosurface value is 0.005 e/Bohr³.

more $H_2$ bubbles are produced per unit area when the same degradation occurs in overpotential, and thus the corresponding catalyst has better stability. As shown in Fig. 4g and Supplementary Table. 5, the $C_{sta}$ of the Mo₂C/MoC/CNT film is as large as $2.57 \times 10^7$ C cm$^{-2}$ V$^{-1}$, which is times or even orders of magnitude higher than the values of other high-performance HER catalysts including 1T-MoS₂, MoNi₄/MoO₂, Ni₂P-Fe₂P, Co-NiS₂, CoP/NiCoP/NC, S-MoS₂@C, single atom NiI, etc. that were measured in alkali, as well as IrFe/NC and Mo₂CTₓ/2H-MoS₂ that were measured in acid[4,8–10,41,44,46–58].

**Mechanism of high catalytic activity and stability**. DFT calculations were performed to further investigate the mechanisms of activity and stability. In general, $\Delta G_{H^*}$ is considered as an important descriptor for analysing the HER activity. If hydrogen binds to the surface weakly, the adsorption step limits the catalytic activity, whereas, if the binding is too strong, the desorption step limits the activity. The $\Delta G_{H^*}$ should be close to zero for an optimal catalyst[1]. In the calculations, oxygen-terminated MoC (111) surface, Mo₂C (100) surface, and Mo₂C/MoC heterostructure were considered for HER, because the existence of high-valence Mo states (Mo⁴⁺, Mo⁵⁺, or Mo⁶⁺) in the XPS spectra (Fig. 2d) suggests partial surface oxidation of Mo₂C, MoC, or Mo₂C/MoC. In addition, the calculated Pourbaix diagram also shows that, under the real reaction condition, the surface of Mo₂C is terminated by oxygen, which plays a key role in the catalytic performance of HER in alkaline media[2]. As shown in Fig. 5a, b, the values of $\Delta G_{H^*}$ for the Mo₂C (100), MoC (111), and Mo₂C (100)/MoC (111) heterostructures were calculated, where β-Mo₂C and α-MoC phases were used for modelling as these two phases

are the main phases as identified by XRD patterns. The values of $\Delta G_{H^*}$ for H and T sites on Mo₂C (100) surface are 0.25 and 0.70 eV, respectively, which shows weak hydrogen adsorption. For the MoC (111) surface, a $\Delta G_{H^*}$ of −0.56 eV indicates strong hydrogen adsorption, which is not favourable for HER. For the Mo₂C/MoC heterostructure, the values of $\Delta G_{H^*}$ for B1, B2, T1, and T2 sites on the Mo₂C/MoC heterostructure are −0.14, −0.16, 0.02, and 0.08 eV, which is much closer to the ideal $\Delta G_{H^*}$ of 0 eV, offering more favourable hydrogen adsorption kinetics toward HER than that of Mo₂C (100) and MoC (111). To reveal why the Mo₂C/MoC interfaces are more active, we performed further DFT calculations to investigate the electronic structures of MoC (111), Mo₂C (100), and Mo₂C/MoC interface. The projected density of states (PDOS) of $p_z$ orbital of O atoms adsorbed on MoC (111), Mo₂C (100), and Mo₂C/MoC interface before and after H adsorption are shown in Supplementary Fig. 37. The $p_z$-band energy center ($\varepsilon_{p_z}$) of the O atoms is investigated to describe the bonding strength of H* on the catalyst surface. The $\varepsilon_{p_z}$ of O atoms on the MoC (111), Mo₂C (100), and Mo₂C/MoC interface are −2.49, −3.86, and −3.09 eV, respectively, which shows that the binding of H* on the oxidized MoC (111) surface is the strongest, while for the oxidized Mo₂C (100) it is the weakest. The binding of H* at the Mo₂C/MoC interface is moderate, which results in excellent thermodynamic activity in HER. These results exhibit a similar trend with the HER activity in the experiment and thus verify that the Mo atoms at the interface are more HER active than Mo³⁺ in MoC or Mo²⁺ in Mo₂C. It is worth noting that the Mo₂C/MoC/CNT film synthesized by the self-heating method has numerous defects, dislocations, and twin boundaries, as shown in the HAADF-STEM image (Supplementary Fig. 38).

DFT calculations show that the absolute values of $\Delta G_{H*}$ for $Mo_2C$ (100) surface with carbon vacancy is almost unchanged compared to those of $Mo_2C$ (100) surface without carbon defects, showing the poor HER activity. However, the absolute value of $\Delta G_{H*}$ for MoC (111) surface with carbon vacancy decreases and is closer to 0 eV than that of MoC (111) surface without carbon defects, showing that the carbon vacancy defects in the MoC (111) surface can improve HER activity. Although carbon vacancy may promote HER activity[57], the $\Delta G_{H*}$ for $Mo_2C/MoC$ interface is as low as 0.02 eV, which is closer to 0 eV than those of $Mo_2C$ (−0.35 or 0.65 eV for (100)) and MoC (−0.04 eV for (111)) surfaces with the carbon vacancy (Supplementary Table. 6 and Fig. 39). In experiments, the HER performance of the $Mo_2C/MoC$ composite is much better than the single $Mo_2C$ or MoC phase. Therefore, in the $Mo_2C/MoC/CNT$ film, we believe that the $Mo_2C/MoC$ interface is the main contribution to the HER activity, and the defects may only additionally promote the HER. The suitable $\Delta G_{H*}$ of $Mo_2C/MoC$ and its unique structure indicate that the interface interaction between $Mo_2C$ and MoC as well as the synthesis by self-heating is favourable for the effective adsorption and activation of the reactant, enhancing the HER performance.

To theoretically study the stability, we establish the heterostructures of $Mo_2C$ (001) or MoC (111) slabs with graphene, instead of CNTs, as a simplified model, as graphene has similar properties to CNTs. The binding energy and charge density differences were calculated, as shown in Fig. 5c. The charges transfer from the $Mo_xC$ slab to graphene and the binding energies for graphene/$Mo_2C$ and graphene/MoC are −62.11 and −52.26 meV/$Å^2$, respectively, which is approximately three times larger than that of bilayer graphene (−23.01 meV/$Å^2$). These results indicate strong coupling between the $Mo_xC$ slab and graphene. Moreover, the shortest distance between $Mo_2C$ (MoC) and graphene is 2.28 (2.25) Å, which is close to the Mo–C bond length of $Mo_2C$ (2.11 Å) (2.17 Å for MoC), indicating that they are bonded to each other.

Moreover, self-heating synthesis is a universal synthesis method suitable for single-phase and composite carbides. As shown in Supplementary Fig. 40, we successfully synthesized well-dispersed single-phase $Nb_4C_3$ and $W_2C/WC$ composite nano-carbides by self-heating within several seconds and regulated the production by adjusting parameters, such as metal/non-metal sources, self-heating time, and output power with high productivity.

## Discussion

In summary, we developed a fast, self-heating (Joule heating) method using a CNT film as heat source and matrix to in-situ synthesize a highly HER active and robust $Mo_2C/MoC/CNT$ film catalyst in the presence of Mo and C precursors. Although Pt-group metal-free and high-current-density HER catalysts are much demanded in practical applications, their fast and scalable synthesis methods have still been very limited. In our method, the heating and cooling processes rapidly occur in hundreds of milliseconds, and the entire synthesis process only lasts for tens of seconds, very advantageous for large-scale and low-cost production. The productivity of the self-heating synthesis reaches about 2000 growth cycles per day, much higher than those of traditional methods such as furnace heating and hydrothermal synthesis.

The as-prepared $Mo_2C/MoC/CNT$ catalyst possesses ultra-low overpotentials of 201, 233, and 255 mV at 500, 1000, and 1500 mA $cm^{-2}$, respectively, in 1 M KOH. The overpotential increases by only 47 mV after working for 14 days at 1000 mA $cm^{-2}$, and changes by less than 10 mV after 10,000 CV cycles. The HER activity of the catalyst even keeps stable at 3000 mA $cm^{-2}$ for days.

Combining the results of experiments and DFT calculations, we reveal the mechanism of the composite catalyst for maintaining both high activity and high stability as follows: 1) The ultrafast heating and cooling and the short growth time facilitate the uniform dispersion of $Mo_2C/MoC$ nanoparticles and the formation of abundant $Mo_2C/MoC$ hetero-interfaces. 2) The charge transfer at the $Mo_2C/MoC$ hetero-interfaces results in the formation of a moderate $H^*$ adsorption of Mo between $Mo^{3+}$ (strong adsorption of $H^*$) and $Mo^{2+}$ (strong desorption of $H^*$) states. Therefore, $Mo_2C/MoC$ hetero-interfaces serve as the main HER active sites with a moderate $\Delta G_{H*}$, in which the adsorption and desorption proceed smoothly even at high current densities. 3) Strong chemical bonds form between $Mo_2C/MoC$ particles and CNT matrix due to the instantaneous high temperature, which prevents the catalyst particles from being peeled off, offering the $Mo_2C/MoC/CNT$ film with long-term stability at high current densities. All of these benefits are difficult to achieve by conventional methods and catalysts. Our study introduces a scalable method to develop electrodes with both high chemical activity and high mechanical stability, which can be applied to the production of industry-scale catalytic electrodes, to ultrafast material synthesis and strengthening, and to many other energy storage applications.

## Methods

**Syntheses**. CNT film was made from super-aligned CNTs (SACNTs), which were synthesized by our previous work through low-pressure chemical vapour deposition[59]. 20 mg SACNTs were dispersed into 250 mL ethanol by high-power probe ultrasonication (SCIENTZ-950E) for 5 min at 600 W. The suspension solution was immediately transferred to the suction filter to form a CNT film. After sufficiently dried in the air, the CNT film was cut into rectangular shapes and drilled with many microscale holes by a direct laser writing machine (1064 nm in wavelength). The drilled holes have a diameter of ~40 mm and a pitch of 800 μm. This kind of holey CNT film effectively releases $H_2$ bubbles during HER, as revealed in our previous work[10].

To synthesize $Mo_2C/MoC/CNT$ film, ammonium molybdate (($NH_4)_2MoO_4·4H_2O$) and glucose ($C_6H_{12}O_6$) with varied atomic ratios of Mo:C (1:1, 4:3, 2:1, and 4:1) were dissolved in a mixed solution of deionized water and ethanol. Urea ($CH_4N_2O$) was then dissolved into the mixed solution based on an optimal atomic ratio of Mo:C (4:3 in the experiment) and varied mole ratios of glucose to urea (0, 10:1, 20:3, 5:1, 5:2). To promote the dissolution of ($NH_4)_2MoO_4·4H_2O$ and prevent the solution from precipitation, we added ammonia water to adjust the pH value of the solution to about 11.5. The precursor was loaded on a CNT film through three times of dip coating and then dried at 60 °C for 10 min in air. The CNT film loaded with precursor was then clamped at both ends by graphite clips and put into a quartz tube. The graphite clips were connected to a stabilized power supply through copper foil. The self-heating process of the CNT film took place in a reductive, mixed atmosphere of 10% $H_2$ and 90% Ar with a certain voltage applied to both ends of the CNT film by a stabilized power supply outputting 30 W or 135 W. The heating time was 30 s at 30 W and ranged from 10 to 120 s at 135 W. The power supply kept an output of 30 W for 30 s at first for carbonization of glucose/urea and decomposition of ammonium molybdate, which prevented $H_2O$ molecules generated by the decomposition of glucose and ammonium molybdate from etching and disintegrating CNTs at high temperatures. Then the output was increased to 135 W and maintained for a short period (10–120 s) to form Mo-C bonds. The as-synthesized $Mo_2C/MoC/CNT$ film is washed with deionized water and ethanol orderly several times and dried before the following electrochemical tests. Besides the self-heating approach to $Mo_2C/MoC/CNT$ film, we also synthesize f-$Mo_2C/MoC/CNT$ films by furnace, with the same pretreatment process but a tube-furnace growth as shown in Supplementary Fig. 25. The loading of the $Mo_2C/MoC$ on the $Mo_2C/MoC/CNT$ film is ~1.8 mg $cm^{-2}$, which indicated ~38 wt % of active material. The thickness is ~0.05 mm and the size is 15 mm × 8 mm.

To synthesize $Mo_2C/CNT$ film, the same pretreatment process was carried out as did in the synthesis of $Mo_2C/MoC/CNT$ film, except that the self-heating growth process was set at 30 W for 20 min. To synthesize MoC/CNT film, the self-heating growth process was set at 30 W for 45 s. The loading of the active materials, i.e., $Mo_2C$ on the $Mo_2C/CNT$ film, and MoC on the MoC/CNT film, are ~2.0 and ~1.7 mg $cm^{-2}$, respectively.

**Characterizations**. Morphology of samples was observed by scanning electron microscope (SEM, ZEISS, Merlin Compact), transmission electron microscopy (TEM, JEOL, JEM-2010F, 200 kV), and spherical correction transmission electron microscope (sc-TEM, JEOL, JEM ARM 200 F, 200 kV). Surface states of the samples were analysed by X-Ray diffraction (XRD, D/max-2500/PC, Rigaku) with Cu Kα radiation (λ = 0.15406 nm) operated at 40 kV and 150 mA. Raman

spectra were collected in the range of 200–800 cm$^{-1}$ with an excitation wavelength of 532 nm (Horiba-iHR550) and X-ray photoelectron spectroscopy (XPS, Thermo Fisher, Escalab 250Xi, Al Kα) was carried out in the range of 0 to 1350 eV at a step of 1 eV. Thermogravimetry (TG, TA INSTRUMENTS, Q5000IR) was performed from 25 to 700 °C in air with a heating rate of 10 °C min$^{-1}$. The temperature of CNT matrix was measured by an infrared thermometer (Optrics, PI640).

**Electrochemical measurements**. All electrochemical measures were performed on CHI 660e and CHI 760e electrochemical workstation by a standard three-electrode test. The CNT composite films directly served as the working electrodes. Hg/HgO served as the reference electrode and Pt as a counter electrode. All the potentials were converted to RHE. The polarization curves were measured at 5 mV s$^{-1}$ with an 85% iR compensation. The cyclic voltammetry curves for fitting double-layer capacitance (C$_{dl}$) were measured at 10–50 mV s$^{-1}$ from 0–0.1 V vs. RHE. The CV cycles for stability test ranged from 0–250 mV with a scan rate of 50 mV s$^{-1}$. Chronopotentiometry (CP, fixed current) was performed at 500, 1000, or 3000 mA cm$^{-2}$ for several days without iR compensation. In order to maintain the pH and liquid level of the electrolyte in the CP test, we introduced a microinjector to continuously replenish deionized water. The noble metal electrodes of Pt/C (20 wt%) were prepared with loading of ~1.9 mg/cm$^{-2}$ on a CNT film via dropping catalyst ink containing Pt/C powder, 50 μL Nafion, 500 μL ethanol, and 450 μL distilled water. Electrochemical impedance spectroscopy (EIS) measurements were tested at an overpotential of 150 mV with a frequency ranging from 10$^5$ to 10$^{-2}$ Hz with an AC amplitude of 5 mV.

**Simulation**. The simulation via COMSOL Multi-Physics software included a geometric model, electric current module, heat transfer module, and electro-magnetic heat module. The geometric model consisted of a CNT film, two copper foils, and four graphite clips with the same dimensions as used in experiments. The conductivity, thermal conductivity and density of CNT were set as 3136 S m$^{-1}$, 28 × (300/T) W m$^{-1}$ K$^{-1}$, and 90 kg m$^{-3}$, respectively. These physical parameters were set according to the experimental parameters, COMSOL database, and literature[60]. When investigating the central temperature of different-length CNT films, the power density was fixed at ~5.125 W mm$^{-3}$.

**Density functional theory calculations**. All calculations based on density functional theory (DFT) were performed using the Vienna ab initio simulation package (VASP)[61]. The projector augmented wave (PAW) potentials[62] and generalized gradient approximation (GGA) of the Perdew-Burke-Ernzerhof (PBE) functional[63] were used for the electron-ion interaction and exchange-correlation energy, respectively. A 3 × 2 × 1 Mo$_2$C (100) slab consisting of six layers of Mo and three layers of C, and a 2 × 2 × 1 MoC (111) slab composed of three layers of Mo and three layers of C were constructed to investigate the HER catalytic performance. Moreover, a 3 × 3 × 1 Mo$_2$C (001) slab consisting of three layers Mo$_2$C and a 2 × 2 × 1 MoC (111) slab were taken into account to establish the heterostructures with a 5 × 5 × 1 graphene supercell. During structural relaxation, the bottom two layers of Mo and C for the Mo$_2$C (100) and MoC (111) and the bottom Mo$_2$C layer for Mo$_2$C (001) slab were fixed. The cutoff energy of the plane wave basis was set to 400 eV. The convergence criteria for the total energy and force were set to 10$^{-5}$ eV and 0.01 eV Å$^{-1}$, respectively. The vacuum layer of at least 15 Å in the z direction was selected to eliminate the interactions between the periodic images. A dipole correction along the z direction of the slab was used in all calculations. The DFT-D3 method[64] was used to describe the Van der Waals interaction. The Gibbs free energy of adsorption hydrogen (ΔG$_{H*}$) is calculated using ΔG$_{H*}$ = ΔE$_{H*}$ + ΔE$_{ZPE}$ − TΔS, where ΔE$_{H*}$, ΔE$_{ZPE}$ and ΔS are the adsorption energy, zero-point energy change and entropy change of hydrogen adsorption, respectively. T is the temperature (T = 298.15 K). Moreover, according to our calculated results, the values of ΔG$_{H*}$ without the implicit solvent environment show the same conclusion as the implicit solvent environment as implemented in VASPsol[65].

# Data availability
The data supporting this study are available within the paper and the Supplementary Information. All other relevant source data are available from the corresponding authors upon reasonable request. Source data are provided with this paper.

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

## Acknowledgements

K.L. acknowledges the financial support from the National Key R&D Program of China (2018YFA0208400), Basic Science Center Project of NSFC under grant No. 51788104, and National Natural Science Foundation of China (51972193). J.L. acknowledges the support from the National Natural Science Foundation of China (11874036), Local Innovative and Research Teams Project of Guangdong Pearl River Talents Program (2017BT01N111), and Basic Research Project of Shenzhen, China (JCYJ20200109142816479, WDZC20200819115243002). Y.S. acknowledges the support from the National Natural Science Foundation of China (11974041).

## Author contributions

K.L. and C.L. conceived the project and designed experiments. C.L., M.L., L.X., and Y.S. fabricated and tested samples. M.L. completed the COMSOL simulation. J.L. and Z.W. performed the DFT calculations. C.L., E.W., and B.W. discussed the working mechanism. C.L., K.L., J.L., Y.S., K.J., and S.F. analysed the data. C.L., K.L., Z.W., J.L., and Y.S. wrote and revised the manuscript. All authors discussed the results and contributed to the final version of the manuscript.

## Competing interests

The authors declare no competing interests.
