## [Peer Review File · Nature Communications]

Ultrafast self-heating synthesis of robust heterogeneous nanocarbides for high current density hydrogen evolution reactionREVIEWER COMMENTS

Reviewer #1 (Remarks to the Author):

Li et al. report a self-heating method to synthesize Mo₂C MXene/MoC/CNT for highly efficient HER with long durability. DFT calculations reveal an almost thermoneutral H* binding energy, optimal for HER activity. The HER performance and stability of the synthesized catalyst is very attractive and one of the highest I have seen, but more work needs to be done to fully characterize (1) the identity of the catalyst formed and (2) the nature of the catalyst interface. I am unsure if the Mo₂C formed is a 2D MXene, and if indeed Mo³⁺ exists at the interface which is responsible for the high HER activity.

Main comments:

- Authors claim that Mo₂C formed is a 2D MXene in the title. This might be true but the authors will have to provide evidence that the Mo₂C sheets formed are 2D (by AFM) and further characterize them. The authors have used Mo₂C nanoparticles instead of Mo₂C MXene reference data for the XRD, Raman, XPS analysis. It is more likely than not that the authors have formed Mo₂C nanoparticles (which they will have to identify which phase it is) and not 2D MXene sheets. Their Mo₂C XRD reference (line 125, Fig 2b) is not Mo₂C MXene (Adv. Funct. Mater. 2016, 26, 3118–3127), and their XRD peak (Fig 2b) will have to show the characteristic (002) MXene peak at low 2θ . The Raman spectrum of Mo₂C in Fig 2a also does not match that of Mo₂C MXene (Chem. Mater. 2017, 29, 6472–6479). The Raman reference for Mo₂C (ref. 23 refers to Mo₂C nanoparticles and not Mo₂C MXene). Notably, Mo₂C MXene has Mo 3d peaks at Mo⁵⁺ and Mo⁶⁺ (Adv. Funct. Mater. 2016, 26, 3118–3127) while the authors have Mo²⁺ peaks in their Mo₂C sample, which suggests that their Mo₂C formed might not be Mo₂C MXene. Mo₂C was also grown within the CNT pores and have small particle sizes (Fig 3c inset), which is in opposition to the main characteristic of MXenes, which are flat large-area 2D materials (Adv. Mater. 2017, 29, 1700072). Fig 3c shows that Mo₂C exists more like a particle than a 2D material, thus calling Mo₂C a MXene might not be appropriate.
- Lines 93-99 and materials section: Authors describe the synthesis procedure using step 1, 2, 3 but the steps are not labelled on Fig 1a. Authors should also mention in which gas environment the films are dried in for step 2, and the gas flow rates for step 3. Authors will need to be more specific in the methods section for readers to reproduce the work: what pH is “alkaline” in line 426, how many times is “several” in line 427/439, and what is “certain proportion” in line 424? It will also help readers if the authors can expand on Fig 1c to include the 30 W and 135 W heating regions and time as a full temperature ramp profile, instead of truncating the profile to show only the rapid heating/cooling at the start/end. Authors can also briefly describe the self-heating phenomena in CNTs in lines 93-99.
- Line 101: for a fair comparison, authors should show the Mo₂C/CNT, MoC/CNT and CNT films before and after the same treatment to show if they are as flexible
- Line 103: authors should specify what is “certain” voltage
- Line 130 and Fig 2c: authors should increase the separation between both graphs as “40” from the left graph is too close to “0.6” from the right graph
- Line 144: authors should provide evidence that Mo³⁺ mixed valency and electron transfer exists solely at the MoC-Mo₂C interface (and not MoC or Mo₂C regions) to justify the lowered Mo⁴⁺ BE and increased Mo²⁺ BE.
- Line 145: authors will need to reason why Mo³⁺ and not Mo²⁺ or Mo⁴⁺ are HER active. It will also help if authors can provide the peak area ratio of Mo²⁺ : Mo³⁺ : Mo⁴⁺ for comparison since the authors will use the XPS data to estimate the CNT : MoC : Mo Also, given that the CNT wt ratio is 61.8%, will it be useful for the authors to reduce the CNT wt % and increase MoC/Mo₂C interfacial area for HER?
- Line 172-173, Fig 3c: does the heating ramp conditions affect the size distribution of the Mo₂C/MoC nanoparticles, beyond the Mo₂C : MoC ratio?
- Line 241-242 and Fig S15: given that MoC/Mo₂C/CNT has an appreciable increase in ECSA compared to MoC/CNT and Mo₂C/CNT, it will be important for the authors to reproduce Fig 4a normalized to ECSA (as an additional SI figure) on top of geometric surface area in Fig 4a. In doing so, the authors might see that the performance of MoC/CNT and Mo₂C/CNT will be closer to

MoC/Mo₂C/CNT. It is also important for the authors to measure the polarization curves of a physical mixture of MoC/CNT + Mo₂C/CNT, since the physical mixture has no Mo³⁺ and MoC/Mo₂C interfaces. Should the MoC/Mo₂C interfacial area be truly important, the physical mixture will not perform as well as MoC/Mo₂C/CNT. Additionally, the authors can calculate the turnover frequency for all 3 variants to illustrate that the activity increase is intrinsic.

- Line 244-246 and supplementary tab 2: the authors should include all the pre- and post-synthesis processing time, including the time it takes to create holes in the CNT. Comparing the time required to synthesize a catalyst might not be entirely fair too, given that different techniques can produce catalysts at different mass scales. The authors can emphasize on their rapid synthesis process, which is a marked improvement from many calcination/slow thermal treatment processes.
- Line 278-281: Authors can try using a graphite counter electrode as a control since they have identified dissolved Pt counter electrode as a possible interference, especially since they are running HER at such high current densities
- Line 297: Authors should give quantitative measures on the intensity of the MoC : Mo₂C peaks after stability testing, just like how they provided in Fig 2c (right)
- Line 322-325 and Fig 4g: Authors will need to verify with all the other papers that the same electrolyte was used for stability testing. It will not be fair to compare HER in alkali (this paper) to HER in acid and especially neutral electrolytes, since the mechanism of HER differs depending on pH.
- Line 356-358: Authors claim that defects and dislocations further enhance HER. What is the proportion of HER activity attributed to defects compared to the MoC/Mo₂C interface?
- Line 372-374: Authors will need to cross-check their synthesized Nb₂C and W₂C compounds against available data for Nb₂C and W₂C MXenes before claiming that they produced 2D MXenes. Similar problem to Mo₂C on the first point.

Minor comments:

- Line 45: Pt-group metals (Ru, Rh, Pd, Ir, Os, Pt) all have very high HER activity and the authors can consider rephrasing non-noble metal as Pt-group metal-free.
- Line 49: Authors suggest that a highly stable catalyst require chemical and mechanical considerations, while a highly active catalyst requires only chemical considerations. By following this statement, wouldn't designing a stable catalyst automatically produce an active catalyst? I would recommend the authors rephrase this sentence for clarity, as there is no opposition in this sentence.
- Line 52-56: citation required to exemplify problem of catalyst exfoliation from HER
- Line 65-66: citation required to exemplify problem of binders obstructing catalytic sites
- Line 74-77: should include "in the presence of Mo and C precursors" to synthesize the film, as CNTs do not provide the Mo or C precursors for film formation
- Line 130 and Fig 2c: authors should increase the separation between both graphs as "40" from the left graph is too close to "0.6" from the right graph
- Lin 155: "By combining XPS with a total weight loss" sounds awkward. It might help to rephrase as "combining XPS and TGA data".
- Line 224: Authors will need to specify 20 wt% of Pt in Pt/C in Fig 4a
- Line 268: Authors are referring to Fig 4e not 4f

Reviewer #2 (Remarks to the Author):

This is an interesting paper on the development of MXene/CNT catalysts for hydrogen evolution. The work has been carried out quite well, the materials have been characterised well and the electroanalysis is reasonably good. I do have reservations about the use of double layer capacitance as a measure of the electrochemical surface area of the materials. The specific double layer capacitance of these materials is not known very accurately and this will lead to a high degree of error in the reported values. Following on from that point, I note there is almost no consideration given to error analysis in the paper. Some estimates of the errors in the reported quantities should be given.

All of the above aside, I am unfortunately not convinced that this paper reveals sufficiently new insights for publication in the journal. There are very many example of materials that are similar to those reported here (differences in preparation methods notwithstanding). I am not convinced that the reported values are of sufficient importance for the readership.

Reviewer #3 (Remarks to the Author):

Title: Ultrafast self-heating synthesis of robust MXene/CNT catalysts for stable high-current-density hydrogen evolution reaction

Recommendation: minor revision

In this paper, CNT was used as matrix and heat source to in situ synthesize Mo₂C/MoC/CNT catalysts for highly active and stable HER process. The ultrafast heating and cooling rate and short growth time benefited the formation of active sites. The Mo₂C/MoC hetero-interface enhanced electron exchange, resulting in the formation of Mo³⁺ sites serve as the main HER active sites. The strong coupling between the Mo₂C and CNT matrix ensure the long-term stability at high current densities of catalysts. Here are some suggestions that were shown below for the improvement of manuscript quality.

1. To improve the electrode stability, the CNT film was drilled by laser for H₂ release channels. Please describe the specific experimental steps of laser drilling.
2. It was mentioned in this paper that Mo³⁺ sites served as the main HER active sites. Please provide sufficient evidence that Mo³⁺ has high HER activity.
3. In the mechanical tensile experiments, whether the improvement of material mechanical strength was related to the rapid rise and fall of joule heating? Please add to prove the comparison sample of pure CNT film heated by joule heating.
4. As shown in Fig. 4e, although the f-Mo₂C/MoC/CNT catalyst deteriorated after 6 days of test, the potential did not change significantly. Please supplement the data of f-Mo₂C/MoC/CNT catalyst tested after 14 days and compare it with the Mo₂C/MoC/CNT catalyst prepared by self-heating.
5. In supplementary Fig. 23, the f-Mo₂C/MoC/CNT catalyst showed the XRD diffraction peak of MoO₃ after 6 days of test, while the Mo₂C/MoC/CNT catalyst prepared by self-heating did not show the MoO₃ peak. Please explain why the self-heating sample has better stability.

Reviewer #1 (Remarks to the Author):

[Comment] Li et al. report a self-heating method to synthesize Mo₂C MXene/MoC/CNT for highly efficient HER with long durability. DFT calculations reveal an almost thermoneutral H* binding energy, optimal for HER activity. *The HER performance and stability of the synthesized catalyst is very attractive and one of the highest I have seen*, but more work needs to be done to fully characterize (1) the identity of the catalyst formed and (2) the nature of the catalyst interface. I am unsure if the Mo₂C formed is a 2D MXene, and if indeed Mo³⁺ exists at the interface which is responsible for the high HER activity.

[Response] We thank the reviewer for all of the comments on our manuscript. As shown in the following responses, we have supplemented more detailed and systematic data in the revised manuscript to make our work more solid. We hope that the reviewer finds our manuscript suitable for publication in Nature Communications now.

[Comment] Main comments:

- Authors claim that Mo₂C formed is a 2D MXene in the title. This might be true but the authors will have to provide evidence that the Mo₂C sheets formed are 2D (by AFM) and further characterize them. The authors have used Mo₂C nanoparticles instead of Mo₂C MXene reference data for the XRD, Raman, XPS analysis. It is more likely than not that the authors have formed Mo₂C nanoparticles (which they will have to identify which phase it is) and not 2D MXene sheets. Their Mo₂C XRD reference (line 125, Fig 2b) is not Mo₂C MXene (Adv. Funct. Mater. 2016, 26, 3118–3127), and their XRD peak (Fig 2b) will have to show the characteristic (002) MXene peak at low 2θ. The Raman spectrum of Mo₂C in Fig 2a also does not match that of Mo₂C MXene (Chem. Mater. 2017, 29, 6472–6479). The Raman reference for Mo₂C (ref. 23 refers to Mo₂C nanoparticles and not Mo₂C MXene). Notably, Mo₂C MXene has Mo 3d peaks at Mo⁵⁺ and Mo⁶⁺ (Adv. Funct. Mater. 2016, 26, 3118–3127) while the authors have Mo²⁺ peaks in their Mo₂C sample, which suggests that their Mo₂C formed might not be Mo₂C MXene. Mo₂C was also grown within the CNT pores and have small particle sizes (Fig 3c inset), which is in opposition to the main characteristic of MXenes, which are flat large-area 2D materials (Adv. Mater. 2017, 29, 1700072). Fig 3c shows that Mo₂C exists more like a particle than a 2D material, thus calling Mo₂C a MXene might not be appropriate.

[Response] We appreciate this valuable comment. Following the reviewer's suggestion, we remeasured the X-ray diffraction (XRD) of the self-heating samples in a wider range of 2θ to further identify the catalyst formed. As reported in the literature (*Nat. Commun.* 12, 5510 (2021); *Adv. Funct. Mater.* 26, 3118-3127 (2016)), there exists an XRD peak of Mo₂C MXene at 2θ ~9° and a c lattice parameter of 20.6 Å. Figure R1 shows that in our sample, there is a weak peak at about 11.5°, corresponding to a reduced c lattice parameter, which is attributed to a transition phase from Mo₂C MXene to β-Mo₂C. This may result from the defunctionalization of the T_x groups (*Nat. Commun.* 12, 5510 (2021); *Chem. Mater.* 31, 4505-4513 (2019)), the removal of intercalated water, and the reestablishment of long-range order in Mo₂C MXene. Because the self-heating synthesis involves an extremely high-temperature process and a reducing H₂ atmosphere, the change of the c lattice parameter is reasonable.

Fig. R1 | XRD spectra of a Mo₂C/MoC/CNT film and a pure CNT film.

In the XRD spectrum of the Mo₂C/MoC/CNT film, other stronger peaks originate from β -Mo₂C and α -MoC (Fig. R1), which indicates that β -Mo₂C and α -MoC are major phases in the composite and the transition phase from Mo₂C MXene to β -Mo₂C is minor. We agree with the reviewer that the Raman spectra, the XPS spectra, and the particle morphology also reveal the dominant β -Mo₂C and α -MoC phases in the composite.

In response to this comment, we added Fig. R1 as new Supplementary Fig. 3, cited the references suggested, revised “MXene/CNT catalyst” into “nano-carbides/CNT catalyst” in the title, and added the above discussion on Page 7 of the revised manuscript.

[Comment] • Lines 93-99 and materials section: Authors describe the synthesis procedure using step 1, 2, 3 but the steps are not labelled on Fig 1a. Authors should also mention in which gas environment the films are dried in for step 2, and the gas flow rates for step 3. Authors will need to be more specific in the methods section for readers to reproduce the work: what pH is “alkaline” in line 426, how many times is “several” in line 427/439, and what is “certain proportion” in line 424? It will also help readers if the authors can expand on Fig 1c to include the 30 W and 135 W heating regions and time as a full temperature ramp profile, instead of truncating the profile to show only the rapid heating/cooling at the start/end. Authors can also briefly describe the self-heating phenomena in CNTs in lines 93-99.

[Response] We thank the reviewer for these comments in details. We respond to these comments point by point as follows.

- (1) We labeled the steps in Fig. 1a of the revised manuscript (also see Fig. R2).

Fig. R2 | a, Schematic illustration of the synthesis of Mo₂C/MoC/CNT films by the self-heating method.

(2) We specified the gas environment in step 2 and the gas flow rates in step 3 on Page 5 of the revised manuscript: “Second, the CNT film loaded with precursors was dried at 60°C for 10 min in air (step 2). Finally, self-heating synthesis was performed in a mixed atmosphere of 10% H₂ and 90% Ar with a total flow rate of 200 sccm, in which the precursors *in situ* reacted on the rapidly Joule-heated CNT film (step 3)”.

(3) We specified the pH of alkaline, the times, and the proportion, etc. in the Methods section of the revised manuscript: “To synthesize Mo₂C/MoC/CNT film, ammonium molybdate ((NH₄)₂MoO₄·4H₂O) and glucose (C₆H₁₂O₆) with varied atomic ratios of Mo:C (1:1, 4:3, 2:1, and 4:1) were dissolved in a mixed solution of deionized water and ethanol. Urea (CH₄N₂O) was then dissolved into the mixed solution based on an optimal atomic ratio of Mo:C (4:3 in the experiment) and varied mole ratios of glucose to urea (0, 10:1, 20:3, 5:1, 5:2). To promote the dissolution of (NH₄)₂MoO₄·4H₂O and prevent the solution from precipitation, we added ammonia water to adjust the pH value of the solution to about 11.5. The precursor was loaded on a CNT film through three times of dip coating and then dried at 60°C for 10 min in air.”

(4) In Fig. 1c, we would like to show the advantage of the rapid heating/cooling in our self-heating method, which allows the control of the entire growth process within a short time. Following the reviewer’s suggestion, we also presented a full temperature ramp profile for a heating process of 30 s at 30 W followed by the other process of 45 s at 135 W, as shown in Fig. R3. The heating and cooling processes in Fig. R3 are also on the order of hundreds of milliseconds. We added this figure as Supplementary Fig. 1a.

Fig. R3 | Temperature control curve of the self-heating process, including a heating process of 30 s at 30 W followed by the other process of 45 s at 135 W.

[Comment] • Line 101: for a fair comparison, authors should show the $\text{Mo}_2\text{C}/\text{CNT}$, MoC/CNT , and CNT films before and after the same treatment to show if they are as flexible

[Response] We thank the reviewer for this valuable comment. As shown in Fig. R4, akin to the $\text{Mo}_2\text{C}/\text{MoC}/\text{CNT}$ film, the pure CNT, MoC/CNT , and $\text{Mo}_2\text{C}/\text{CNT}$ films also remain flexible before and after the rapid treatment at ~ 1770 K. We added Fig. R4 as new Supplementary Fig. 2 and the above description on Page 5 of the revised manuscript.

Fig. R4 | **a-b**, Optical images showing the good flexibility of a pure CNT film and a precursor/CNT film before a self-heating process. **c-f**, Optical images of a pure CNT film, MoC/CNT film, $\text{Mo}_2\text{C}/\text{CNT}$ film, and $\text{Mo}_2\text{C}/\text{MoC}/\text{CNT}$ film, respectively, after the self-heating process. All of the films keep flexible before or after the treatment.

[Comment] • Line 103: authors should specify what is “certain” voltage

[Response] The threshold voltage per unit length applied for the CNT films emitting visible light is ~ 0.5 V/mm. We have specified this voltage value on Page 5 of the revised manuscript.

[Comment] • Line 130 and Fig 2c: authors should increase the separation between both graphs as “40” from the left graph is too close to “0.6” from the right graph.

[Response] We have separated the graphs in Fig. 2c to avoid the crowding of both x-axes. The new Fig. 2c is shown as follows.

Fig. R5 | XRD spectra of Mo₂C/MoC/CNT films synthesized with different heating time at 135W (left panel) and the corresponding peak intensity ratios of MoC (111) to Mo₂C (002) (right panel). The error bars represent standard deviations.

[Comment] • Line 144: authors should provide evidence that Mo³⁺ mixed valency and electron transfer exists solely at the MoC-Mo₂C interface (and not MoC or Mo₂C regions) to justify the lowered Mo⁴⁺ BE and increased Mo²⁺ BE.

[Response] We thank the reviewer for this valuable comment. Following the reviewer’s suggestion, we provided more experimental and theoretical evidence to support the electron transfer scenario at the Mo₂C/MoC interface.

First, we carefully reviewed the valence states of Mo in Mo₂C and MoC reported in the literature and re-analyzed the change of valence states in our materials. It is still debating for the exact valence states of Mo in Mo₂C or MoC. Most studies suggest the dominance of Mo²⁺ in Mo₂C and Mo³⁺ in MoC, and higher valence states of Mo⁴⁺, Mo⁵⁺, and Mo⁶⁺ result from partial oxidation (*Nat. Catal.* 1, 960-967 (2018); *Nat. Commun.* 12, 6776 (2021); *J. Am. Chem. Soc.* 140, 14481-14489 (2018)). Under this assumption, we re-analyzed the XPS spectra. As shown in Fig. R6, compared with the Mo²⁺ peak in Mo₂C and the Mo³⁺ peak in MoC, in the Mo₂C/MoC/CNT film the Mo²⁺ peak apparently blueshifts while the Mo³⁺ peak redshifts, suggesting the existence of electron transfer from Mo₂C to MoC in the heterogeneous composite. Note that the strong Mo⁴⁺, Mo⁵⁺, and Mo⁶⁺ peaks in the MoC sample should result from the surface MoO_x because MoC is very prone to oxidation.

Fig. R6 | XPS spectra of Mo 3d in MoC powder (top), Mo₂C/MoC/CNT film (middle), and Mo₂C powder (bottom).

Second, we investigated the charge distribution in Mo₂C and MoC by DFT calculations. As shown in Fig. R7, each Mo atom loses about -0.80 and -0.50 electron, respectively, in the body of the MoC (111) and Mo₂C (100) regions, while at the Mo₂C/MoC interface, each Mo atom loses about -0.75 and -0.70 electron because of the electron transfer from Mo₂C to MoC regions. This result clearly proves that the charge transfer exists solely at the Mo₂C/MoC interface.

Fig. R7 | Charge transfer at the Mo₂C/MoC interface revealed by DFT calculations.

In response to this comment, we added Fig. R6 and R7 as new Fig. 2d and Supplementary Fig. 8. We also added the above response on Page 8 of the revised manuscript.

[Comment] • Line 145: authors will need to reason why Mo³⁺ and not Mo²⁺ or Mo⁴⁺ are HER active. It will also help if authors can provide the peak area ratio of Mo²⁺ : Mo³⁺ : Mo⁴⁺ for comparison since the authors will use the XPS data to estimate the CNT : MoC : Mo. Also, given that the CNT wt ratio is 61.8%, will it be useful for the authors to reduce the CNT wt % and increase MoC/Mo₂C interfacial area for HER?

[Response] We thank the reviewer for this valuable comment. As we re-analysed in the last response, we note that the dominant valence state of Mo is +2 in Mo₂C and +3 in MoC mainly. The exact valence state of Mo at the Mo₂C/MoC interface may be between +2 and +3 due to the electron transfer over there. Under such a scenario, it is not possible to use the peak area ratio of Mo²⁺: Mo³⁺: Mo⁴⁺ as an indicator to compare the HER activity.

To reveal why the Mo₂C/MoC interface are more active, we performed further DFT calculations to investigate the electronic structures of MoC (111), Mo₂C (100), and Mo₂C/MoC interface. The projected density of states (PDOS) of *p_z* orbital of O atoms adsorbed on MoC (111), Mo₂C (100), and Mo₂C/MoC interface before and after H adsorption are shown in Fig. R8. The *p*-band center (ϵ_p) of the O atoms is investigated to describe the bonding strength of H on the catalyst surface. The ϵ_p of O atoms on the MoC (111), Mo₂C (100), and Mo₂C/MoC interface are -2.49, -3.86, and -3.09 eV, respectively, which shows that the binding of H on the oxidized MoC (111) surface is the strongest, while for the oxidized Mo₂C (100) it is the weakest. The binding of H at the Mo₂C/MoC interface is moderate. According to the Sabatier principle, the adsorption of H on a catalyst should not be too weak or too strong. Weak adsorption would lower the capability of hydrogen formation and strong adsorption would restrict the release of hydrogen. The moderate adsorption of H at the Mo₂C/MoC interface could result in the excellent thermodynamic activity in HER. These results are consistent with the trend of the HER activity in the experiment and thus verify that the Mo at the interface are more HER active than Mo³⁺ in MoC or Mo²⁺ in Mo₂C.

Fig. R8 | Projected density of states (PDOS) of O-*p_z* orbital of O atoms adsorbed on MoC (111) surface, Mo₂C (100) surface, and Mo₂C/MoC interface before and after H adsorption. The solid red line indicates the *p*-band center (ϵ_p) of *p_z* orbital of O atoms. The Fermi level is set to 0 eV.

We also varied the CNT wt% by loading different amounts of precursors on CNT films. The CNT wt% in the Mo₂C/MoC/CNT samples prepared from precursors loaded by 1, 3, and 5 times is about 83.0%, 61.8%, and 49.3%, respectively. As shown in Fig. R9, the sample with the moderate CNT wt% (~ 61.8 wt% CNT and ~ 38.2 wt% Mo₂C/MoC) possesses the best HER activity. It exhibits no apparent agglomeration of nanoparticles and thus has abundant Mo₂C/MoC interfacial area (Fig.

R10b). More or less content of Mo₂C/MoC interfaces will lead to poorer HER activity. When the content of CNT is high, the content of MoC or Mo₂C is very few and thereby the total Mo₂C/MoC interfacial area becomes very limited (Fig. R10a). And for the samples with a low content of CNT, Mo₂C/MoC will agglomerate during the self-heating process, also reducing the total MoC/Mo₂C interfacial area (Fig. R10c). This result suggests that an appropriate CNT wt% is needed to increase the Mo₂C/MoC interfacial area for a better HER.

Fig. R9 | Polarization curves of Mo₂C/MoC/CNT films with different CNT wt%.

Fig. R10 | SEM images of Mo₂C/MoC/CNT films with different CNT wt%.

In response to this comment, we added Fig. R8, R9, and R10 as new Supplementary Fig. 37, Supplementary Fig. 17, and Supplementary Fig. 18, respectively. We also added the above discussion on Page 13 and 22 of the revised manuscript.

[Comment] • Line 172-173, Fig 3c: does the heating ramp conditions affect the size distribution of the Mo₂C/MoC nanoparticles, beyond the Mo₂C : MoC ratio?

[Response] We thank the reviewer for this valuable comment. The heating ramp conditions do affect the size distribution of the Mo₂C/MoC nanoparticles. To show that, we changed the heating ramp time for both heating steps from room temperature to ~ 1100 K (30 W) and from ~ 1100 to ~1770 K (135 W). As the heating ramp rate decreases, the particles gradually agglomerate and increase in size because they are more likely to diffuse and aggregate during the heating process, especially at high temperatures (Fig. R11).

Fig. R11 | a-c SEM images of as-prepared Mo₂C/MoC/CNT films at different heating ramp rates. The heating ramp time for both heating steps from room temperature to ~ 1100 K (30 W) and from ~ 1100 to ~1770 K (135 W) is 10 s, 200 s, and 500 s, respectively.

In response to this comment, we added Fig. R11 as Supplementary Fig. 10, and the above response on Page 10 of the revised manuscript.

[Comment] • Line 241-242 and Fig S15: given that MoC/Mo₂C/CNT has an appreciable increase in ECSA compared to MoC/CNT and Mo₂C/CNT, it will be important for the authors to reproduce Fig 4a normalized to ECSA (as an additional SI figure) on top of geometric surface area in Fig 4a. In doing so, the authors might see that the performance of MoC/CNT and Mo₂C/CNT will be closer to MoC/Mo₂C/CNT. It is also important for the authors to measure the polarization curves of a physical mixture of MoC/CNT + Mo₂C/CNT, since the physical mixture has no Mo³⁺ and MoC/Mo₂C interfaces. Should the MoC/Mo₂C interfacial area be truly important, the physical mixture will not perform as well as MoC/Mo₂C/CNT. Additionally, the authors can calculate the turnover frequency for all 3 variants to illustrate that the activity increase is intrinsic.

[Response] We thank the reviewer for these comments. We supplemented more experiments and analyses to show the ECSA-normalized polarization curve, the effect of the physical mixture, and the turnover frequency.

First, the ECSA of each sample can be evaluated from the double-layer capacitance (C_{dl}) according to

$$ECSA = \frac{C_{dl}}{C_s},$$

where C_s is the specific capacitance of the sample or the capacitance of an atomically smooth planar surface of the material per unit area under the same condition. C_s for a flat surface is generally found to be in the range of 20-60 $\mu\text{F cm}^{-2}$ (*Nat. Energy* 4, 512-518 (2019); *Nat. Mater.* 18, 1309-1314 (2019)), and the value of 40 $\mu\text{F cm}^{-2}$ is adopted in this work to calculate the ECSA.

The ECSAs of Mo₂C/MoC/CNT film, Mo₂C/CNT film, and MoC/CNT film are calculated to be 2998, 2088, and 1650 cm^2 , respectively, according to previous CV results. The polarization curves of three films normalized to ECSA are re-plotted in Fig. R12, which demonstrates that the activity of Mo₂C/MoC/CNT film does increase intrinsically compared with those of Mo₂C/CNT film and MoC/CNT film.

Fig. R12 | Polarization curves of Mo₂C/MoC/CNT film, Mo₂C/CNT film, and MoC/CNT film, normalized by ECSA.

Second, we found that it was very difficult to disperse Mo₂C/CNT film and MoC/CNT film and physically mix them. Instead, we mechanically mixed the MoC and Mo₂C powders, dispersed them in the ethanol/water/Nafion solution, and dropped the solution onto a CNT film to prepare a physically mixed Mo₂C/MoC/CNT film (m-Mo₂C/MoC/CNT film). Because of the lack of the chemical bonding interaction and the efficient charge transfer between Mo₂C and MoC, the m-Mo₂C/MoC/CNT film has a much poorer HER performance than the Mo₂C/MoC/CNT film prepared by the self-heating process (Fig. R13), which elucidates that the Mo₂C/MoC interface is truly important.

Fig. R13 | Polarization curves of a Mo₂C/MoC/CNT film prepared by the self-heating process (red) and a physically mixed Mo₂C/MoC/CNT film made by loading mixed MoC and Mo₂C powders onto the CNT film (cyan).

Moreover, we also evaluated the turnover frequency (TOF) of each catalyst film (Fig. R14). The TOF is calculated by the following equation (*Nat. Energy* 4, 512-518 (2019); *Nat. Commun.* 10, 3755 (2019)):

$$\text{TOF}(\text{s}^{-1}) = \frac{|j| \text{ (mA cm}^{-2}\text{)}}{n \times 1000 \times N \times \text{ECSA} \times (1.602 \times 10^{-19} \text{C})} = \frac{3.12 \times 10^{15} \times |j|}{N \times \text{ECSA}}$$

where N is the density of active sites, n is the number of electrons involved in the reaction. The density of active sites can be calculated as follows,

$$N = \left(\frac{\text{Number of sites/unit cell}}{\text{Volume/unit cell}} \right)^{\frac{2}{3}}$$

For MoC,

$$N_1 = \left(\frac{8 \text{ sites/unit cell}}{77.854 \text{ \AA}^3/\text{unit cell}} \right)^{\frac{2}{3}} = 2.19 \times 10^{15} \text{ sites cm}^{-2}$$

For Mo₂C,

$$N_2 = \left(\frac{4 \text{ sites/unit cell}}{37.459 \text{ \AA}^3/\text{unit cell}} \right)^{\frac{2}{3}} = 2.25 \times 10^{15} \text{ sites cm}^{-2}$$

For MoC/Mo₂C,

$$N_3 = xN_1 + yN_2 = 2.22 \times 10^{15} \text{ sites cm}^{-2},$$

where x=49%, y=51% according to the XPS results.

At an overpotential of 250 mV, the TOF of Mo₂C/MoC/CNT film, Mo₂C/CNT film, and MoC/CNT film is 0.65 s⁻¹, 0.30 s⁻¹, and 0.22 s⁻¹, respectively, which validates that Mo₂C/MoC/CNT catalyst has higher intrinsic activity besides the larger ECSA. The increase of the intrinsic activity should be attributed to the abundant Mo₂C/MoC interfaces.

Fig. R14 | Turnover frequency per surface Mo atom (TOF_{Mo}) of Mo₂C/MoC/CNT film, Mo₂C/CNT film, and MoC/CNT film.

In response to this comment, we added Fig. R12, R13, and R14 as Supplementary Fig. 23, Supplementary Fig. 19, and Supplementary Fig. 24, respectively, and the above response on Page 14 and 15 of the revised manuscript.

[Comment] • Line 244-246 and supplementary tab 2: the authors should include all the pre- and post-synthesis processing time, including the time it takes to create holes in the CNT. Comparing the time required to synthesize a catalyst might not be entirely fair too, given that different techniques can produce catalysts at different mass scales. The authors can emphasize on their rapid synthesis process, which is a marked improvement from many calcination/slow thermal treatment processes.

[Response] We thank the reviewer for this valuable comment. In our method, the pre-synthesis processing includes CNT film preparation, laser drilling, and precursor loading, which takes about 15 minutes in total. After a self-heating synthesis at high temperature, the as-prepared composite film can be served as an electrode directly. Our method costs a comparable pre-/post-synthesis processing time and a much shorter synthesis time than the traditional methods. However, since the authors did not clearly describe the time required for pre- and post-processing using other methods in the literature, it is difficult to directly compare the time required to synthesize a catalyst, including all the pre- and post-synthesis processing time.

In response to the comment, we listed the “synthesis time” excluding the pre- and post-synthesis processing time in Supplementary Table 2, and noted the pre-/post-synthesis processing time is ~15 minutes below the table. We described the table on Page 15 of the revised manuscript as follows: “Besides the excellent HER activity of the Mo₂C/MoC/CNT film at high current densities, our self-heating method has notable advantages in the rapid synthesis process and high productivity, which takes a comparable pre-/post-synthesis processing time and a much shorter synthesis time than traditional methods (Fig. 4b and Supplementary Tab. 2).”

[Comment] • Line 278-281: Authors can try using a graphite counter electrode as a control since they have identified dissolved Pt counter electrode as a possible interference, especially since they are running HER at such high current densities

[Response] We thank the reviewer for this suggestion. We tried to use a graphite counter electrode (Gaoss Union) in our stability measurement under a large current density. However, the graphite counter electrode was found to be dissolved in the KOH solution after the test at a current density of 1000 mA cm⁻² for ~4 days, causing the interruption of the measurement (Fig. R15). In spite of this, the unchanged overpotential during the first 4 days still reveals the high stability of our Mo₂C/MoC/CNT film.

Fig. R15 | Long-term stability at 1000 mA cm⁻² measured using a graphite rod as the counter electrode in 1M KOH.

Actually, many previous studies used Pt as the counter electrode in the high-current-density HER

(*Nat. Energy* 4, 107-114 (2018); *Nat. Mater.* 18, 1309-1314 (2019); *Nat. Commun.* 9, 2609 (2018)). In our experiment, after a long-term test at a high current density, the content of Pt in the electrolyte was below the detection limit of inductively coupled plasma (ICP) mass spectrometry, which excludes the influence of Pt dissolution from counter electrodes during the electrochemical measurements (Supplementary Tab. 4).

We also used a graphite rod as the counter electrode to measure the CV curves of Mo₂C/MoC/CNT films for 50 cycles, and then changed to use a Pt counter electrode for the other 50-cycles CV measurement. The CV curves obtained by graphite and Pt counter electrodes are identical (Fig. R16), also verifying that the Pt counter electrode does not influence the results.

Fig. R16 | CV curves measured using a graphite counter rod for the first 50 cycles and then a Pt counter electrode for the other 50 cycles.

In summary, we found that graphite rods are not suitable as counter electrodes in long-term tests at high current density. Therefore, we used Pt instead of graphite rods as the counter electrodes for the stability measurement. In response to this comment, we added Fig. R15 and R16 as Supplementary Fig. 28 and Fig. 29, respectively, and the above response on Page 17 of the revised manuscript.

[Comment] • Line 297: Authors should give quantitative measures on the intensity of the MoC : Mo₂C peaks after stability testing, just like how they provided in Fig 2c (right)

[Response] We thank the reviewer for this valuable comment. We quantitatively measured the XRD peak intensity of MoC (111) and Mo₂C (101) after the stability test according to Supplementary Fig. 33. The peak intensity ratio is MoC (111) : Mo₂C (101) ~ 1.4 after the test, which is higher than its pristine value (~ 0.7) (Fig. R17). The reason may lie in the fact that Mo₂C (101) is more susceptible to corrosion than MoC (111). Although the material is corroded, the remained excellent HER performance supports our statement that the Mo₂C/MoC interface, rather than single MoC or Mo₂C, is crucial in the HER activity.

In response to this comment, we added Fig. R17 as Supplementary Fig. 33b and the above discussion on Page 18 of the revised manuscript.

Fig. R17 | Peak intensity ratio of MoC (111) to Mo₂C (101) changes from ~0.7 for the pristine Mo₂C/MoC/CNT film to ~1.4 for the Mo₂C/MoC/CNT film working at 1000 mA cm⁻² for ~6 days.

[Comment] • Line 322-325 and Fig 4g: Authors will need to verify with all the other papers that the same electrolyte was used for stability testing. It will not be fair to compare HER in alkali (this paper) to HER in acid and especially neutral electrolytes, since the mechanism of HER differs depending on pH.

[Response] We thank the reviewer for this suggestion. In Fig. 4g, almost all catalysts for comparison are measured in alkali with pH=14. We agree with the reviewer that the mechanism is different in alkali/acid/neutral electrolytes, but high stability of the catalytic electrode is required regardless of the electrolyte type.

In response to this comment, we marked the type of electrolytes in Fig. 4g (also see Fig. R18), and revised the description for the comparison on Page 20 as follows, “As shown in Fig. 4g and Supplementary Tab. 5, the C_{sta} of the Mo₂C/MoC/CNT film is as large as $2.57 \times 10^7 \text{ C cm}^{-2} \text{ V}^{-1}$, which is times or even orders of magnitude higher than the values of other high-performance HER catalysts including 1T-MoS₂, MoNi₄/MoO₂, Ni₂P-Fe₂P, Co-NiS₂, CoP/NiCoP/NC, S-MoS₂@C, single atom NiI, etc. that were measured in alkali, as well as IrFe/NC and Mo₂CT_x/2H-MoS₂ that were measured in acid.”

Fig. R18 | Comparison of long-term stability of various HER catalysts at both small and high current

densities.

[Comment] • Line 356-358: Authors claim that defects and dislocations further enhance HER. What is the proportion of HER activity attributed to defects compared to the MoC/Mo₂C interface?

[Response] We thank the reviewer for this valuable comment. By using DFT calculations, we investigate the vacancy formation energies of Mo and C. We reveal that C vacancy is easier to form than Mo vacancy for both Mo₂C and MoC surfaces (Table R1). Thus, we further investigate the ΔG_{H^*} of the Mo₂C (100), MoC (111) surfaces with C vacancy, as shown in Fig. R19. DFT calculations show that the absolute values of ΔG_{H^*} for Mo₂C (100) surface with C vacancy are almost unchanged compared to those of Mo₂C (100) surface without C defects, showing the poor HER activity. However, the absolute value of ΔG_{H^*} for MoC (111) surface with C vacancy decreases and is closer to 0 eV than that of MoC (111) surface without C defects, showing that the carbon vacancy defects in the MoC (111) surface can improve HER activity.

Although C vacancy may promote HER activity, the ΔG_{H^*} for Mo₂C/MoC interface is as low as 0.02 eV, which is closer to 0 eV than that of Mo₂C (-0.35 or 0.65 eV for (100)) and MoC (-0.04 eV for (111)) surfaces with the C vacancy. In experiments, the HER performance of the Mo₂C/MoC composite is much better than the single Mo₂C or MoC phase. Therefore, in the Mo₂C/MoC/CNT film, we believe that the Mo₂C/MoC interface is the main contribution to the HER activity, and the defects may only additionally promote the HER activity.

In response to this comment, we added Table R1 and Fig. R19 as Supplementary Tab. 6 and Fig. 39. We also added the above discussion on Page 22 and 23 of the revised manuscript and in Supplementary Information.

Table R1 | Vacancy formation energies of Mo or C in Mo₂C and MoC surfaces.

Surface type	Mo vacancy (eV)	C vacancy (eV)
Mo ₂ C	2.71	0.35
MoC	1.87	-1.39

Fig. R19 | Adsorption structures and ΔG_{H^*} of hydrogen on the Mo₂C (100) and MoC (111) surface with the carbon vacancy.

[Comment] • Line 372-374: Authors will need to cross-check their synthesized Nb₂C and W₂C

compounds against available data for Nb₂C and W₂C MXenes before claiming that they produced 2D MXenes. Similar problem to Mo₂C on the first point.

[Response] We thank the reviewer for this valuable comment. Similar to the analysis of Mo₂C/MoC/CNT film as mentioned above, Nb₄C₃ and W₂C/WC composite contains mainly 3D-lattice structures (Fig. R20, also as Supplementary Fig. 40). For rigorous expression, we described them as nano-carbides in the revised manuscript.

Fig. R20 | **a**, Raman spectra of a Nb₄C₃/CNT film. **b**, XRD pattern of the Nb₄C₃/CNT film. **c**, SEM image of the Nb₄C₃/CNT film. **d**, EDS mapping of the Nb element in the Nb₄C₃/CNT film and the corresponding SEM image (inset). **e**, Raman spectra of a W₂C/WC/CNT film. **f**, XRD pattern of the W₂C/WC/CNT film. **g**, SEM image of the W₂C/WC/CNT film. **h**, EDS mapping of the W element in the W₂C/WC/CNT film and the corresponding SEM image (inset).

[Comment] Minor comments:

• Line 45: Pt-group metals (Ru, Rh, Pd, Ir, Os, Pt) all have very high HER activity and the authors can consider rephrasing non-noble metal as Pt-group metal-free.

[Response] We thank the reviewer for this valuable comment. Following the suggestion, we revised the description of “non-noble metal” as “Pt-group metal-free” throughout the manuscript.

[Comment] • Line 49: Authors suggest that a highly stable catalyst require chemical and mechanical considerations, while a highly active catalyst requires only chemical considerations. By following this statement, wouldn't designing a stable catalyst automatically produce an active catalyst? I would recommend the authors rephrase this sentence for clarity, as there is no opposition in this sentence.

[Response] We thank the reviewer for this valuable comment. We rephrase this sentence on Page 3 of the revised manuscript as follows: “However, the development of high-efficiency and Pt-group metal-free HER catalytic electrodes for high-current-density HER is challenging, because it requires simultaneous high chemical activity, high chemical stability, and high mechanical stability of the electrodes.”

[Comment] • Line 52-56: citation required to exemplify problem of catalyst exfoliation from HER

[Response] We cited the relevant literature (*Nat. Commun.* **12**, 6051 (2021); *Nat. Energy* **2**, 17172 (2017); *Adv. Mater.* **26**, 2683-2687 (2014)) as Ref. 14, 15, and 16 to exemplify the problem of catalyst exfoliation.

[Comment] • Line 65-66: citation required to exemplify problem of binders obstructing catalytic sites

[Response] We cited the relevant literature (*Nat. Commun.* **12**, 6051 (2021); *Nat. Commun.* **9**, 3132 (2018); *Adv. Mater.* **31**, e1808167 (2019)) as Ref. 14, 23, and 24 to exemplify the problem of binders.

[Comment] • Line 74-77: should include “in the presence of Mo and C precursors” to synthesize the film, as CNTs do not provide the Mo or C precursors for film formation

[Response] We revised this sentence into “we develop a low-energy-consumption method using a carbon nanotube (CNT) film as a heat source and matrix, which rapidly changes its temperature in hundreds of milliseconds to *in situ* synthesize a robust Mo₂C/MoC/CNT composite film in the presence of Mo and C precursors.” on Page 4 of the revised manuscript.

[Comment] • Line 130 and Fig 2c: authors should increase the separation between both graphs as “40” from the left graph is too close to “0.6” from the right graph

[Response] We adjusted the separation in Fig. 2c as suggested.

[Comment] • Lin 155: “By combining XPS with a total weight loss” sounds awkward. It might help to rephrase as “combining XPS and TGA data”.

[Response] Following the suggestion, we rephrased this description into “combining XPS and TGA data”.

[Comment] • Line 224: Authors will need to specify 20 wt% of Pt in Pt/C in Fig 4a

[Response] We specified 20 wt% of Pt in Pt/C in the caption of Fig. 4a.

[Comment] • Line 268: Authors are referring to Fig 4e not 4f

[Response] We thank the reviewer for his/her carefulness. We have corrected this typo.

Reviewer #2 (Remarks to the Author):

[Comment] This is an interesting paper on the development of MXene/CNT catalysts for hydrogen evolution. *The work has been carried out quite well, the materials have been characterised well and the electroanalysis is reasonably good.* I do have reservations about the use of double layer capacitance as a measure of the electrochemical surface area of the materials. The specific double layer capacitance of these materials is not known very accurately and this will lead to a high degree of error in the reported values. Following on from that point, I note there is almost no consideration given to error analysis in the paper. Some estimates of the errors in the reported quantities should be given.

[Response] We appreciate the reviewer's evaluation and valuable comments on our manuscript. In the revised manuscript, we have supplemented more detailed and systematic data to make our work more solid, and further highlighted the novelty of our work.

In response to the specific comment, we would like to note that the electrochemical surface area (ECSA), which is proportional to the double-layer capacitance (C_{dl}), is usually used to estimate the density of active sites (*Nat. Commun.* 11, 5462 (2020); *Sci. Adv.* 5, eaav6009 (2019)). The double-layer capacitance (C_{dl}) is obtained by fitting the relationship between the variation of currents and the sweeping rate at a fixed potential in the non-Faradaic region, because in this region almost all currents come from the capacitance effect. This method is widely adopted in other works (for example, *Nat. Energy* 4, 512-518 (2019); *Nat. Mater.* 18, 1309-1314 (2019)).

The ECSA of a material can be calculated from the double-layer capacitance (C_{dl}) by the following equation,

$$ECSA = \frac{C_{dl}}{C_s},$$

where C_s is the specific capacitance of the material or the capacitance of an atomically flat surface per unit area under the same condition.

For C_{dl} , the measurement error includes systematic errors (0.01% for voltage application and 0.2% for current detection) and random error (~5%), which is negligible. For C_s , its value for a flat surface is generally found to be in the range of 20-60 $\mu\text{F cm}^{-2}$ (*Nat. Energy* 4, 512-518 (2019); *Nat. Mater.* 18, 1309-1314 (2019)), and the value of 40 $\mu\text{F cm}^{-2}$ is adopted in this work to calculate the ECSA. As the actual C_s cannot be determined precisely in this work, this treatment may cause an error in the absolute value of ECSA. However, the relative values of the ECSA for the materials reported in this work are not affected by the absolute error of C_s , because the $\text{Mo}_2\text{C}/\text{MoC}/\text{CNT}$ film, $\text{Mo}_2\text{C}/\text{CNT}$ film, and MoC/CNT film are prepared on the same supporting material, the CNT films. The C_s values for all these materials should be nearly identical in principle, and thus the calculated ECSAs can be compared relatively. With the specific C_s of 40 $\mu\text{F cm}^{-2}$, the ECSAs of $\text{Mo}_2\text{C}/\text{MoC}/\text{CNT}$ film, $\text{Mo}_2\text{C}/\text{CNT}$ film, and MoC/CNT film are calculated to be 2998, 2088, and 1650 cm^2 , respectively. The polarization curves of the three films are re-plotted in Fig. R12, which demonstrates that the activity of $\text{Mo}_2\text{C}/\text{MoC}/\text{CNT}$ film does increase intrinsically compared with that of $\text{Mo}_2\text{C}/\text{CNT}$ film and MoC/CNT film.

In response to this comment, we have added Fig. R12 as Supplementary Fig. 23, and part of the above discussion on Page 15 of the revised manuscript.

Fig. R12 | Polarization curves of Mo₂C/MoC/CNT film, Mo₂C/CNT film, and MoC/CNT film normalized by ECSA.

[Comment] All of the above aside, I am unfortunately not convinced that this paper reveals sufficiently new insights for publication in the journal. There are very many example of materials that are similar to those reported here (differences in preparation methods notwithstanding). I am not convinced that the reported values are of sufficient importance for the readership.

[Response] We thank the reviewer for the comment. It is well-known that the synthesis methods are indeed crucial for the high activity and robustness of the noble metal-free catalysts for high-current-density HER. Although MoC and Mo₂C were investigated for high-current-density HER catalysts in recent years, the synthesis of an HER catalyst that can work at high current densities (≥ 1000 mA cm⁻²) for weeks is still challenging. As highlighted by Reviewer #1, our synthesized catalyst possesses the almost highest performance in HER activity and stability among the results reported. And to our knowledge, this work is the first to show an HER catalyst that can be working at 1000 mA/cm² for two weeks without noticeable degradation. This excellent HER performance benefits from the superfast self-heating synthesis, which not only produces a highly active heterogeneous electrocatalyst but also enhances its stability.

In the following, we will summarize the advantage of self-heating synthesis and highlight the novelty of our work.

(1) **Extremely high productivity of synthesis.** Although noble-metal-free and high-current-density HER catalysts are much demanded in practical applications (*Nat. Mater.* 18, 1309-1314 (2019); *Nat. Commun.* 11, 2940 (2020)), their fast and scalable synthesis methods have still been very limited. In this work, we develop a fast, self-heating (Joule heating) method using the CNT film as heat source and matrix to *in-situ* synthesize a robust Mo₂C/MoC/CNT HER catalyst in the

presence of Mo and C precursors. In this method, the heating and cooling processes rapidly occur in hundreds of milliseconds, and the entire synthesis process only lasts for tens of seconds. The productivity of the self-heating synthesis reaches about **2,000 growth cycles per day**, much higher than those of traditional methods such as furnace heating and hydrothermal synthesis.

(2) **Novel mechanism for both high activity and high stability.** For the improvement of the stability of the HER electrocatalyst, binders would be almost inevitably referred to (*Nat Commun* 10, 631 (2019); *Sci. Adv.* 5, eaav6009 (2019)). However, binders would weaken the chemical activity of catalysts and may still fail at high current densities due to corrosion. In our method, owing to the extremely high temperature in the self-heating synthesis, the as-prepared, uniformly dispersed Mo₂C/MoC heterogeneous nanoparticles form strong chemical bonds with the CNT heater/electrode. The numerous Mo₂C/MoC hetero-interfaces offer abundant active sites with a moderate hydrogen adsorption free energy ΔG_H (0.02 eV), and the strong chemical bonding significantly enhances the mechanical stability of the Mo₂C/MoC/CNT catalyst. Therefore, high activity and high stability can be achieved simultaneously in our catalyst.

(3) **Excellent HER performance at high current densities.** Due to the heterogeneous compositing and strong chemical bonding, the Mo₂C/MoC/CNT catalyst possesses an ultra-low overpotential of 255 mV at 1500 mA cm⁻² in 1 M KOH. The overpotential shows only a very slight change after the catalyst works at 1000 mA cm⁻² for 14 days. The HER activity of the catalyst even keeps stable at 3000 mA cm⁻² for days. The high activity and high stability suggest that our non-noble-metal HER catalyst has excellent HER performance, even better than those ever reported in the literature, to our knowledge. The promising HER activity, excellent stability, and high productivity of our catalyst can fulfill the demands of hydrogen production in practical applications.

We believe that our work represents a breakthrough in the synthetic and mechanism innovation of novel heterogeneous catalysts for high-current-density HER applications. In response to this comment, we have added part of the above response in the Discussion section on Page 24, 25 and 26 in the revised manuscript to highlight the novelty of our work. We hope that the reviewer finds our manuscript suitable for publication in Nature Communications now.

Reviewer #3 (Remarks to the Author):

[Comment] Title: Ultrafast self-heating synthesis of robust MXene/CNT catalysts for stable high-current-density hydrogen evolution reaction

Recommendation: minor revision

In this paper, CNT was used as matrix and heat source to in situ synthesize Mo₂C/MoC/CNT catalysts for highly active and stable HER process. The ultrafast heating and cooling rate and short growth time benefited the formation of active sites. The Mo₂C/MoC hetero-interface enhanced electron exchange, resulting in the formation of Mo³⁺ sites serve as the main HER active sites. The strong coupling between the Mo_xC and CNT matrix ensure the long-term stability at high current densities of catalysts. Here are some suggestions that were shown below for the improvement of manuscript quality.

[Response] We thank the reviewer for the evaluation of our manuscript and all of the comments. As shown in the following responses, we have supplemented more experiments and theoretical calculations in the revised manuscript to make our conclusion more solid. We hope that the reviewer finds our manuscript suitable for publication in Nature Communications now.

[Comment] 1. To improve the electrode stability, the CNT film was drilled by laser for H₂ release channels. Please describe the specific experimental steps of laser drilling.

[Response] We thank the reviewer for this valuable comment. The CNT films were drilled with many microscale holes by a direct laser writing machine (1064 nm in wavelength). The drilled holes have a diameter of ~ 40 μm and a pitch of 800 μm. This kind of holey CNT film effectively releases H₂ bubbles during HER, as revealed in our previous work (*J. Mater. Chem. A*, 8, 17527 (2020)). We have added the above description on Page 26 of the revised manuscript.

[Comment] 2. It was mentioned in this paper that Mo³⁺ sites served as the main HER active sites. Please provide sufficient evidence that Mo³⁺ has high HER activity.

[Response] We thank the reviewer for this valuable comment. We carefully reviewed the valence states of Mo in Mo₂C and MoC reported in the literature and re-analyzed the change of valence states in our materials. It is still debating for the exact valence states of Mo in Mo₂C or MoC. Most studies suggest the dominance of Mo²⁺ in Mo₂C and Mo³⁺ in MoC, and higher valence states of Mo⁴⁺, Mo⁵⁺, and Mo⁶⁺ result from partial oxidation (*Nat. Catal.* 1, 960-967 (2018); *Nat. Commun.* 12, 6776 (2021); *J. Am. Chem. Soc.* 140, 14481-14489 (2018)). The exact valence state of Mo at the Mo₂C/MoC interface may be between +2 and +3 due to the electron transfer over there.

To reveal why the MoC/Mo₂C interface are more active, we performed further DFT calculations to investigate the electronic structures of MoC (111), Mo₂C (100), and MoC/Mo₂C interface. The projected density of states (PDOS) of *p_z* orbital of O atoms adsorbed on MoC (111), Mo₂C (100), and Mo₂C/MoC interface before and after H adsorption are shown in Fig. R8. The *p*-band center (ϵ_p)

of the O atoms is investigated to describe the bonding strength of H on the catalyst surface. The ϵ_p of O atoms on the MoC (111), Mo₂C (100), and Mo₂C/MoC interface are -2.49, -3.86, and -3.09 eV, respectively, which shows that the binding of H on the oxidized MoC (111) surface is the strongest, while for the oxidized Mo₂C (100) it is the weakest. The binding of H at the Mo₂C/MoC interface is moderate. According to the Sabatier principle, the adsorption of H on a catalyst should not be too weak or too strong. Weak adsorption would lower the capability of hydrogen formation and strong adsorption would restrict the release of hydrogen. The moderate adsorption of H at the Mo₂C/MoC interface could result in excellent thermodynamic activity in HER. These results are consistent with the trend of the HER activity in the experiment and thus verify that the Mo at the interface are more HER active than Mo³⁺ in MoC or Mo²⁺ in Mo₂C.

Fig. R8 | Projected density of states (PDOS) of O- p_z orbital of O atoms adsorbed on MoC (111) surface, Mo₂C (100) surface, and Mo₂C/MoC interface before and after H adsorption. The solid red line indicates the p -band center (ϵ_p) of p_z orbital of O atoms. The Fermi level is set to 0 eV.

In response to this comment, we added Fig. R8 as Supplementary Fig. 37 and the above response on Page 22 of the revised manuscript.

[Comment] 3. In the mechanical tensile experiments, whether the improvement of material mechanical strength was related to the rapid rise and fall of joule heating? Please add to prove the comparison sample of pure CNT film heated by joule heating.

[Response] We thank the reviewer for the valuable comment. We measured the mechanical properties of a pure CNT film before and after the self-heating process. As shown in Fig. R21, after the self-heating, the breaking strain of the pure CNT film is notably reduced while the breaking strength only slightly increases to ~0.32 MPa. In contrast, the Mo₂C/MoC/CNT film synthesized by the self-heating process has a much higher breaking strength (~6.87 MPa). These results suggest that the enhancement of the mechanical strength of the Mo₂C/MoC/CNT film should be mostly attributed to the strong interaction between Mo₂C/MoC and CNTs, which improves the load transfer efficiency inside the film.

In response to this comment, we added Fig. R21 as Supplementary Fig. 36b and the above

discussion on Page 19 of the revised manuscript.

Fig. R21 | Stress-strain curves of a pure CNT film before and after a self-heating process (30W for 30s and then 135W for 45s).

[Comment] 4. As shown in Fig. 4e, although the f-Mo₂C/MoC/CNT catalyst deteriorated after 6 days of test, the potential did not change significantly. Please supplement the data of f-Mo₂C/MoC/CNT catalyst tested after 14 days and compare it with the Mo₂C/MoC/CNT catalyst prepared by self-heating.

[Response] We thank the reviewer for this valuable comment. Following the reviewer's suggestions, we tested the long-term stability of an f-Mo₂C/MoC/CNT film to compare it with the Mo₂C/MoC/CNT film by self-heating. As shown in Fig. R22, the overpotential of the Mo₂C/MoC/CNT film increases by only ~47 mV after working at 1000 mA cm⁻² for 14 days (336 h), while the overpotential of the f-Mo₂C/MoC/CNT film increases by ~344 mV, more than 7 times that of the Mo₂C/MoC/CNT film, during the same testing period (14 days). Therefore, the stability of f-Mo₂C/MoC/CNT film is much inferior to that of the Mo₂C/MoC/CNT film, as we claimed in the manuscript.

Fig. R22 | Long-term stability of self-heating and furnace-heating samples at 500 or 1000 mA cm⁻² without iR compensation.

In response to this comment, we replaced Fig. 4e by Fig. R22 and added the above discussion on Page 17 of the revised manuscript.

[Comment] 5. In supplementary Fig. 23, the f-Mo₂C/MoC/CNT catalyst showed the XRD diffraction peak of MoO₃ after 6 days of test, while the Mo₂C/MoC/CNT catalyst prepared by self-heating did not show the MoO₃ peak. Please explain why the self-heating sample has better stability.

[Response] We thank the reviewer for this valuable comment. The Mo₂C/MoC/CNT catalyst has better resistance to oxidation during HER because of its higher crystallinity than f-Mo₂C/MoC/CNT film (see Fig. 2b vs Supplementary Fig. 25b and Supplementary Fig. 31 a-b vs Supplementary Fig. 31 c-d). In the fast self-heating synthesis, the growth temperature is very high (~1770 K), which is beneficial for the high crystallinity. In the furnace-heating, however, it is difficult to obtain high crystallinity of the f-Mo₂C/MoC/CNT film and simultaneously maintain the Mo₂C/MoC composite phase due to the much longer heating process at a lower temperature. Therefore, f-Mo₂C/MoC/CNT film has a lower crystallinity and will be gradually oxidized during the high-current-density HER.

In response to this comment, we added part of the above discussion on Page 18 of the revised manuscript.

REVIEWER COMMENTS

Reviewer #1 (Remarks to the Author):

The authors have significantly improved their manuscript and addressed the primary concerns raised: (1) the phase of the Mo₂C formed, (2) missing details in the experimental section, (3) the chemical nature of the interface and its importance in explaining the high HER activity and (4) providing additional negative controls to showcase the HER activity. Overall, the manuscript is improved greatly and will be ready for publication after the minor comments below are addressed. I appreciate the authors' efforts in addressing the individual pointers raised previously.

Regarding (1), supplementary figure 3 is a very important complement to Fig 2b, which illustrates clearly that the Mo₂C is primarily α and β phase, and not as a MXene, as the authors have clarified. This is because the 2D Mo₂C MXene (002) peak is very weak and very broad, which is uncharacteristic of 2D layered materials with a well-defined c-lattice parameter. In this regard, I suggest that the authors provide a simple quantitative estimate of the α to β phase ratio through Le Bail XRD peak fitting, which will be very useful for readers to know. A minor comment here will be to initially first refer to MXene as 2D Mo₂C, as this paper is expected to be read by colleagues in the bulk transition metal carbide ceramics field (α and β phase), who might (or might not) be familiar with 2D transition metals and carbides (MXenes). It will also be useful for authors to replace "Tx" with surface terminations since Tx is a MXene terminology and is now hardly referred to in this manuscript. Nonetheless, the authors have very clearly identified the main phases of Mo₂C which is very important for future benchmarking.

On (2), the experimental section is now more adequate for readers to reproduce the data.

For (3), the new analysis of the Mo 3d XPS spectrum and the additional charge distribution calculation does indeed suggest that the MoC/Mo₂C interface is directly responsible for the HER activity. The p band center argument agrees with the conclusions of the ΔG calculations in showing thermoneutrality for H* binding, although I am unsure if the p band centers of the three scenarios indeed fall on the apex of the volcano plot as the authors suggest (to illustrate that the interface's intermediate p band position is optimal). A minor comment here is to label what element each colored sphere represents in Figure R7. In the calculations for Fig 5, can the authors specify which phase is used for modelling Mo₂C, and if it is consistent with the main phase identified after XRD peak fitting?

With regards to (4) the negative controls provides stronger evidence that the authors' materials are indeed intrinsically more active, when normalized to ECSA and when considering intrinsic performance indicators like TOF. Can I check if the HER polarization curves in Figure R9 is normalized to the mass of the active material Mo₂C/MoC present?

Reviewer #2 (Remarks to the Author):

The authors have written an extensive response to all comments by the reviewers. I cannot confidently comment on the authors' response to Reviewer 1's concerns about whether or not an MXene has been formed. I think that the publishers must refer to Reviewer 1 for an updated take on whether he/she is confident in the material characterisation.

When it comes to the response to my concerns, I note that the authors have defended the use of the specific capacitance to measure the electrochemical surface area. I do realise that this has been done many times, and is generally accepted. I just think that it comes with some caveats. The accepted value is 20-60 microfarads per square centimetre and generally researchers use as a value of 40. I

accept that this is generally done but I have reservations about it. The actual value could be very different from that reported throughout the paper and I would suggest that some acknowledgment of this would add to the paper.

Overall, I still have some reservations about the potential impact of the work. Having said that, as long as the authors have satisfactorily addressed the comments of reviewers 1 and 2 (to their satisfaction) I think the paper can probably be accepted.

Reviewer #3 (Remarks to the Author):

No comments, it should be accepted.

Reviewer #4 (Remarks to the Author):

Comments to "Ultrafast self-heating synthesis of robust nano-carbides/CNT catalysts for stable high-current-density hydrogen evolution reaction", by Chenyu Li et al

This is a joint theory-experimental work discussing high current density for HER by carbides/CNT catalysis. The efficiency and stability are quite impressive. Given my expertise I will only comment on the theory part. I recommend minor revision before proceeding:

1. It is difficult to understand O pz energy center at interfaces' relation to HER activity. I understand its connection to ORR or OER; why this is important for H adsorption energy at surface in HER? The authors suddenly introduced oxidized surface but that is a different system from the ones which the authors are studying in this work. This is quite confusing.
2. The energy scales the authors care about for H binding energy are very small, e.g. 0.02 eV, but vibrational entropy and solvation energy were not explicitly included. This can introduce over 0.1 or 0.2 eV error already. Can we believe the results at this energy scale?
3. The interface structure is complicated: there is intrinsic lattice mismatch 5-7% from the lattice constant; how did the authors make the interface, e.g. stretch one or compress another? Is this important for H binding energy?
4. The authors mentioned about adding a layer of graphene on top of this interface. Will HER happen on top of graphene layer at the end? Or happen at positions without graphene layer? This was confusing too.

Overall the authors can do a better job of explaining the theory details and how to justify their models.

REVIEWER COMMENTS

Reviewer #1 (Remarks to the Author):

[Comment] The authors have significantly improved their manuscript and addressed the primary concerns raised: (1) the phase of the Mo₂C formed, (2) missing details in the experimental section, (3) the chemical nature of the interface and its importance in explaining the high HER activity and (4) providing additional negative controls to showcase the HER activity. Overall, the manuscript is improved greatly and will be ready for publication after the minor comments below are addressed. I appreciate the authors' efforts in addressing the individual pointers raised previously.

[Response] We thank the reviewer for the positive evaluation on our manuscript and all of the comments. As shown in the following responses, we have supplemented more analyses in the revised manuscript. We hope that the reviewer finds our manuscript suitable for publication in Nature Communications now.

[Comment] Regarding (1), supplementary figure 3 is a very important complement to Fig 2b, which illustrates clearly that the Mo₂C is primarily α and β phase, and not as a MXene, as the authors have clarified. This is because the 2D Mo₂C MXene (002) peak is very weak and very broad, which is uncharacteristic of 2D layered materials with a well-defined c-lattice parameter. In this regard, I suggest that the authors provide a simple quantitative estimate of the α to β phase ratio through Le Bail XRD peak fitting, which will be very useful for readers to know.

[Response] We appreciate this valuable comment. Following the reviewer's suggestion, we quantitatively estimate the content of α -MoC and β -Mo₂C by Le Bail and Rietveld XRD peak fitting of the data (Fig. 2b) in the analysis software GSAS. We first imported the crystallographic information files (cif) of α -MoC and β -Mo₂C as the reference phases for Le Bail extraction, and then used the Rietveld refinement for further analysis of α -MoC-to- β -Mo₂C phase ratio. As shown in Fig. R1, I(obs) is the original data and I(calc) is the Rietveld refined result, both of which fit quite well and the R_{wp} is 7.1%. After the analysis in GSAS, the fitted weight percentages of α -MoC and β -Mo₂C are 59.8% and 40.2%, respectively, corresponding to an α -MoC-to- β -Mo₂C molar ratio of about 2.8:1.

Fig. R1 | XRD peak fitting of Mo₂C/MoC/CNT film by Le Bail and Rietveld refinement in the

analysis software GSAS.

According to the molar content of α -MoC to β -Mo₂C obtained from the Le Bail and Rietveld refinement, we replotted the turnover frequency (TOF) curves, as shown in Fig. R2. At an overpotential of 250 mV, the TOFs of Mo₂C/MoC/CNT film, Mo₂C/CNT film, and MoC/CNT film are 0.65, 0.30, and 0.22 s⁻¹, respectively. These values are identical to the previous TOFs based on the XPS fitting, which validates the highest intrinsic activity of the Mo₂C/MoC/CNT catalyst and does not change the conclusion.

Fig. R2 | Turnover frequency per surface Mo atom (TOF_{Mo}) of Mo₂C/MoC/CNT film, Mo₂C/CNT film, and MoC/CNT film.

In response to this comment, we added Figs. R1 and R2 as Supplementary Fig. 3a and Supplementary Fig. 24, respectively. We also added the above description on Page 7 of the revised manuscript.

[Comment] A minor comment here will be to initially first refer to MXene as 2D Mo₂C, as this paper is expected to be read by colleagues in the bulk transition metal carbide ceramics field (α and β phase), who might (or might not) be familiar with 2D transition metals and carbides (MXenes). It will also be useful for authors to replace “T_x” with surface terminations since T_x is a MXene terminology and is now hardly referred to in this manuscript. Nonetheless, the authors have very clearly identified the main phases of Mo₂C which is very important for future benchmarking.

[Response] We thank the reviewer for these good suggestions. In our revised manuscript, we have referred to MXene as 2D Mo₂C, and also replaced “T_x” with the expression of “surface terminations”.

[Comment] On (2), the experimental section is now more adequate for readers to reproduce the data.

[Response] We thank the reviewer for the positive evaluation.

[Comment] For (3), the new analysis of the Mo 3d XPS spectrum and the additional charge distribution calculation does indeed suggest that the MoC/Mo₂C interface is directly responsible for the HER activity. The p band center argument agrees with the conclusions of the ΔG calculations in showing thermoneutrality for H* binding, although I am unsure if the p band centers of the three scenarios indeed fall on the apex of the volcano plot as the authors suggest (to illustrate that the interface's intermediate p band position is optimal).

[Response] We thank the reviewer for this comment. The p_z -band energy center of the oxygen (ϵ_{p_z}) could be considered as a good descriptor for the hydrogen adsorption free energy (ΔG_{H^*}). According to DFT calculations, the values of ΔG_{H^*} (ϵ_{p_z}) for the MoC (111) surface, the Mo₂C (100) surface, and the Mo₂C/MoC interface are 0.70 (-3.86), -0.56 (-2.49), and 0.02 (-3.09) eV, respectively. Fig. R3 shows that the values of ΔG_{H^*} have a linear relationship with the values of ϵ_{p_z} of the MoC (111) surface, the Mo₂C (100) surface, and the Mo₂C/MoC interface. Therefore, the ϵ_{p_z} of the MoC (111) surface should be located to the left of the apex of the volcano (strong hydrogen adsorption), and the ϵ_{p_z} of the Mo₂C (100) surface should be located to the right of the apex of the volcano (weak hydrogen adsorption), while the ϵ_{p_z} of the Mo₂C/MoC interface should be located near the apex of the volcano.

Fig. R3 | Linear relationship between the hydrogen adsorption free energy (ΔG_{H^*}) and p_z -band energy center of the oxygen (ϵ_{p_z}) on the MoC (111) surface, Mo₂C (100) surface, and Mo₂C/MoC interface.

[Comment] A minor comment here is to label what element each colored sphere represents in Figure R7.

[Response] We appreciate this comment. As shown in Fig. R4, we labelled the element of Mo and C the colored spheres represent in this figure (re-numbered as Supplementary Fig. 8).

Fig. R4 | Charge transfer at the Mo₂C/MoC interface revealed by DFT calculations.

[Comment] In the calculations for Fig 5, can the authors specify which phase is used for modelling Mo₂C, and if it is consistent with the main phase identified after XRD peak fitting?

[Response] We thank the reviewer for this valuable comment. We used β -Mo₂C (JCPDS No. 35-0787) and α -MoC (JCPDS No. 89-2868) for modelling. These two phases are consistent with the main phases we have identified by XRD patterns. In response to this comment, we specified the phases on Page 22 of the revised manuscript.

[Comment] With regards to (4) the negative controls provides stronger evidence that the authors' materials are indeed intrinsically more active, when normalized to ECSA and when considering intrinsic performance indicators like TOF. Can I check if the HER polarization curves in Figure R9 is normalized to the mass of the active material Mo₂C/MoC present?

[Response] We thank the reviewer for the positive evaluation. In the original Fig. R9, the mass of the active materials, i.e., Mo₂C/MoC on the Mo₂C/MoC/CNT film, Mo₂C on the Mo₂C/CNT film, and MoC on the MoC/CNT film, are 2.1, 2.4, and 2.0 mg, respectively, which have only a little difference. The HER polarization curves shown in the last round of responses to comments had not been normalized to the mass. Following the reviewer's suggestion, we normalized the polarization curves to the mass of the active materials in the three kinds of films (Fig. R5), which also shows the highest activity of the Mo₂C/MoC/CNT film and does not change the conclusion.

Fig. R5 | Polarization curves of Mo₂C/MoC/CNT film, Mo₂C/CNT film, and MoC/CNT film, normalized by the mass of their active materials (i.e., Mo₂C/MoC, Mo₂C, and MoC, respectively).

In response to this comment, we added Fig. R5 as Supplementary Fig. 23b. We also added the above description on Page 15 of the revised manuscript.

Reviewer #2 (Remarks to the Author):

[Comment] The authors have written an extensive response to all comments by the reviewers. I cannot confidently comment on the authors' response to Reviewer 1's concerns about whether or not an MXene has been formed. I think that the publishers must refer to Reviewer 1 for an updated take on whether he/she is confident in the material characterisation. When it comes to the response to my concerns, I note that the authors have defended the use of the specific capacitance to measure the electrochemical surface area. I do realise that this has been done many times, and is generally accepted. I just think that it comes with some caveats. The accepted value is 20-60 microfarads per square centimetre and generally researchers use as a value of 40. I accept that this is generally done but I have reservations about it. The actual value could be very different from that reported throughout the paper and I would suggest that some acknowledgment of this would add to the paper. Overall, I still have some reservations about the potential impact of the work. Having said that, as long as the authors have satisfactorily addressed the comments of reviewers 1 and 2 (to their satisfaction) I think the paper can probably be accepted.

[Response] We appreciate the reviewer's evaluation and all of the valuable comments on our manuscript. We also thank the reviewer for her/his acceptance of the use of the specific capacitance to evaluate the ECSA in our work. Following the reviewer's suggestion, we added some acknowledgement of the difference between the actual value and the reported value of the specific capacitance on Page 15 of the revised manuscript as follows, "C_s for a flat surface is generally found to be in a range of 20-60 $\mu\text{F cm}^{-2}$, and the value of 40 $\mu\text{F cm}^{-2}$ is used in this work to calculate the ECSA. The actual value of C_s could be very different from the used value and thus an error may be introduced in the absolute value of ECSA."

As the reviewer will see in this round of comments, previous Reviewer #3 has already suggested publication of our manuscript, and previous Reviewer #1 recommended publication after some minor comments are addressed. The new Reviewer #4, also recommended minor revision and only commented on the theory part of our manuscript. We believe that our work will have potential impact for the high-current-density HER applications due to the high productivity of materials synthesis and the synergistic mechanism of heterogeneous electrocatalysts. We hope that the reviewer finds our revised manuscript suitable for publication in Nature Communications now.

Reviewer #3 (Remarks to the Author):

[Comment] No comments, it should be accepted.

[Response] We appreciate the reviewer's evaluation and all of the comments she/he raised during the review process.

Reviewer #4 (Remarks to the Author):

[Comment] This is a joint theory-experimental work discussing high current density for HER by carbides/CNT catalysis. The efficiency and stability are quite impressive. Given my expertise I will only comment on the theory part. I recommend minor revision before proceeding:

[Response] We thank the reviewer for the positive evaluation on our manuscript and all the comments. As shown in the following responses, we have supplemented more analyses in the revised manuscript to make our work more solid. We hope that the reviewer finds our revised manuscript suitable for publication in Nature Communications now.

[Comment] It is difficult to understand O p_z energy center at interfaces' relation to HER activity. I understand its connection to ORR or OER; why this is important for H adsorption energy at surface in HER? The authors suddenly introduced oxidized surface but that is a different system from the ones which the authors are studying in this work. This is quite confusing.

[Response] We thank the reviewer for this comment. The p_z -band energy center of the oxygen (ϵ_{p_z}) could be considered as a good descriptor for the hydrogen adsorption free energy (ΔG_{H^*}). According to the DFT calculations, the values of ΔG_{H^*} (ϵ_{p_z}) for the MoC (111) surface, the Mo₂C (100) surface, and the Mo₂C/MoC interface are 0.70 (−3.86), −0.56 (−2.49), and 0.02 (−3.09) eV, respectively. Fig. R3 shows that the values of ΔG_{H^*} have a linear relationship with the values of ϵ_{p_z} of the MoC (111) surface, the Mo₂C (100) surface, and the Mo₂C/MoC interface.

Fig. R3 | Linear relationship between the hydrogen adsorption free energy (ΔG_{H^*}) and p_z -band energy center of the oxygen (ϵ_{p_z}) on the MoC (111) surface, the Mo₂C (100) surface, and the Mo₂C/MoC interface.

As discussed in our manuscript, there appear high-valence Mo states (Mo⁴⁺, Mo⁵⁺, or Mo⁶⁺) in the XPS spectra (Fig. 2d), which should originate from MoO_x due to the partial surface oxidation of Mo₂C, MoC, or Mo₂C/MoC. This result indicates that either Mo₂C, MoC, or Mo₂C/MoC heterostructure is subject to surface oxidation, and thereby their surfaces are probably oxygen-terminated. In addition, the calculated Pourbaix diagram also shows that, under the real reaction condition, the surface of Mo₂C is terminated by oxygen, which plays a key role in the catalytic

performance of HER in alkaline media (*Nat. Commun.* 2019, 10, 269). According to the above results, the oxygen-terminated MoC (111) surface, Mo₂C (100) surface, and Mo₂C/MoC heterostructure are considered for HER in our study.

In response to this comment, we added the above discussion on Page 22 of the revised manuscript.

[Comment] The energy scales the authors care about for H binding energy are very small, e.g. 0.02 eV, but vibrational entropy and solvation energy were not explicitly included. This can introduce over 0.1 or 0.2 eV error already. Can we believe the results at this energy scale?

[Response] We thank the reviewer for this valuable comment. In our manuscript, the Gibbs free energy of adsorption hydrogen (ΔG_{H^*}) was used to evaluate the HER activity of the catalyst. ΔG_{H^*} was calculated using $\Delta G_{H^*} = \Delta E_{H^*} + \Delta E_{ZPE} - T\Delta S$, where ΔE_{H^*} , ΔE_{ZPE} and ΔS are the adsorption energy, zero-point energy change, and entropy change of hydrogen adsorption, respectively. Therefore, the vibrational entropy has been included in our study. According to the reviewer's suggestion, the values of ΔG_{H^*} for MoC (111) surface, Mo₂C (100) surface, and Mo₂C/MoC interface with the implicit solvent environment as implemented in VASPsol were also calculated and shown in Table R1. For comparison, the values of ΔG_{H^*} without the implicit solvent environment are also shown in Table R1. The difference in ΔG_{H^*} is in a range of ± 0.1 eV with the inclusion of the implicit solvent environment. However, according to the values of ΔG_{H^*} with the solvent effect, the Mo₂C/MoC interface also shows excellent thermodynamic activity in HER, which does not change our conclusion in the manuscript.

In response to this comment, we added the above discussion on Page 31 of the revised manuscript.

Table R1 | Gibbs free energy of adsorption hydrogen (ΔG_{H^*}) for MoC (111) surface, Mo₂C (100) surface, and Mo₂C/MoC heterostructure without and with the implicit solvent environment.

	Adsorption sites	ΔG_{H^*} without implicit solvent (eV)	ΔG_{H^*} with implicit solvent (eV)	Difference (eV)
MoC (111)	T	-0.56	-0.64	-0.08
	vac	-0.04	-0.13	-0.09
Mo ₂ C (100)	H	0.25	0.31	+0.06
	T	0.70	0.75	+0.05
	vac1	-0.35	-0.29	+0.06
	vac2	0.65	0.63	-0.02
Mo ₂ C/MoC Interface	B1	-0.14	-0.03	+0.09
	B2	-0.17	-0.16	+0.01
	T1	0.02	-0.09	-0.11
	T2	0.08	0.12	+0.04

[Comment] The interface structure is complicated: there is intrinsic lattice mismatch 5-7% from the lattice constant; how did the authors make the interface, e.g. stretch one or compress another? Is this important for H binding energy?

[Response] We thank the reviewer for this valuable comment. According to experimental results, MoC is cubic structure ($a = b = c = 4.35 \text{ \AA}$) and Mo₂C is hexagonal structure ($a = b = 3.04 \text{ \AA}$, $c = 4.63 \text{ \AA}$). The lattice constant of the MoC (111) surface is $a = b = 6.15 \text{ \AA}$, while the lattice constants of Mo₂C (100) surface are $a = 3.04 \text{ \AA}$ and $b = 4.63 \text{ \AA}$. To minimize the lattice mismatch between the MoC (111) surface and the Mo₂C (100) surface, we constructed the Mo₂C/MoC heterostructure by using the ($2 \times \sqrt{3}$) supercell of MoC (111) surface and (4×2) supercell of Mo₂C (100) surface. Therefore, at the interface, the lattice constants of the MoC (111) surface and the Mo₂C (100) surface are 12.30 and 12.16 \AA , respectively. The mismatch is only 1.1%. In our manuscript, the lattice constant of the (4×2) supercell of Mo₂C (100) surface was stretched from 12.16 to 12.30 \AA , and then the Mo₂C/MoC heterostructure ($a = 12.30 \text{ \AA}$) was constructed. Moreover, according to the reviewer's suggestion, the Mo₂C/MoC heterostructure, constructed by compressing the lattice constant of the ($2 \times \sqrt{3}$) supercell of MoC (111) surface (from 12.30 to 12.16 \AA), was also utilized to investigate the Gibbs free energy of adsorption hydrogen (ΔG_{H^*}). As shown in Table R2, the values of ΔG_{H^*} for Mo₂C/MoC heterostructure with the compression of MoC (111) surface are almost unchanged.

Table R2 | Gibbs free energy of adsorption hydrogen (ΔG_{H^*}) on Mo₂C/MoC interface with the different lattice constants of the Mo₂C/MoC heterostructure.

Adsorption sites	ΔG_{H^*} with $a = 12.30 \text{ \AA}$ (eV)	ΔG_{H^*} with $a = 12.16 \text{ \AA}$ (eV)
B1	-0.14	-0.16
B2	-0.17	-0.15
T1	0.02	0.01
T2	0.08	0.05

[Comment] The authors mentioned about adding a layer of graphene on top of this interface. Will HER happen on top of graphene layer at the end? Or happen at positions without graphene layer? This was confusing too.

[Response] We apologize for the confusing description of the effect of graphene, which may make the reviewer misunderstand. In the DFT calculations, graphene was used to study the possible chemical bonding between the catalyst and the CNT substrate, rather than to study the activity of the catalyst. The reason for using graphene instead of CNTs is to simplify the model, as graphene has similar properties to CNTs. The binding energies between Mo₂C (MoC) and graphene were calculated, indicating that they could be bonded to each other, which results in the high mechanical stability of the Mo₂C/MoC/CNT catalysts. However, for the activity of HER, we calculated the Gibbs free energies of adsorption hydrogen (ΔG_{H^*}) of MoC (111) surface, Mo₂C (100) surface, and Mo₂C/MoC interface without the inclusion of the graphene. In real reaction, HER happens at the

catalyst active positions exposed to the electrolyte and the CNTs serve as a supporting and conductive substrate.

In response to this comment, we specified the role of graphene on Page 24 of the revised manuscript.

REVIEWERS' COMMENTS

Reviewer #1 (Remarks to the Author):

The authors have characterized their materials adequately, including the phases of the Mo₂C catalyst. Negative controls after normalizing catalytic data and turnover also supports their hypothesis. From an experimental point of view, the paper is acceptable for publication and I thank the authors for the additional experiments and clarifications to strengthen their manuscript. The current manuscript can be accepted as is, after reviewer #4 evaluates the theoretical aspects of this manuscript.

Reviewer #4 (Remarks to the Author):

Comments to "Ultrafast self-heating synthesis of robust nano-carbides/CNT catalysts for stable high-current-density hydrogen evolution reaction"

I have carefully checked the revised manuscript and responses. The authors have answered my questions and addressed my concerns successfully. I agree with publishing.

REVIEWER COMMENTS

Reviewer #1 (Remarks to the Author):

[Comment] The authors have characterized their materials adequately, including the phases of the Mo₂C catalyst. Negative controls after normalizing catalytic data and turnover also supports their hypothesis. From an experimental point of view, the paper is acceptable for publication and I thank the authors for the additional experiments and clarifications to strengthen their manuscript. The current manuscript can be accepted as is, after reviewer #4 evaluates the theoretical aspects of this manuscript.

[Response] We appreciate the reviewer's evaluation and all of the insightful comments she/he raised during the review process. As the reviewer will see in this round of comments, previous Reviewer #4 has already suggested publication of our manuscript. We are glad that our revised version of the manuscript satisfies the reviewer.

Reviewer #4 (Remarks to the Author):

[Comment] I have carefully checked the revised manuscript and responses. The authors have answered my questions and addressed my concerns successfully. I agree with publishing.

[Response] We appreciate the reviewer's evaluation and all of the comments she/he raised during the review process.